# Development of bridge failure model and fragility curves for infrastructure overturning and deck sliding due to lahars

Joaquín Dagá[1], Alondra Chamorro[2], Hernán de Solminihac[3], Tomás Echaveguren[4]

[1]Master of Engineering Sciences, Department of Engineering and Construction Management, Faculty of Engineering, Pontificia Universidad Católica de Chile, Santiago, Chile.

[2]Associate Professor, School of Engineering, Pontificia Universidad Católica de Chile. Research Associate, National Research Center for Integrated Disaster Risk Management (CIGIDEN), Santiago, Chile.

[3]Professor, School of Engineering, Pontificia Universidad Católica de Chile. Director of the Engineering and Construction Management Department, School of Engineering, Pontificia Universidad Católica de Chile. Director of the Latin American Center of Economic and Social Policies UC (CLAPES UC), Santiago, Chile.

[4]Associate Professor, Civil Engineering Department, Faculty of Engineering, Universidad de Concepción. Researcher, National Research Center for Integrated Disaster Risk Management (CIGIDEN), Concepción, Chile.

*Correspondence to*: Joaquín Dagá (jadaga@uc.cl)

**Abstract.** One of the main volcanic processes affecting road infrastructure are lahars, which are flows of water and volcanic material running down the slopes of a volcano and river valleys. In this paper, a model of bridge failure due to lahars is proposed and, based on this, fragility curves for infrastructure overturning and deck sliding are developed. The failure model considers the limit state of the infrastructure overturning moment and the tangential force over the deck caused by lahars. Analytical models to estimate these loads were calibrated to simulate the effect of lahars over bridges for the development of fragility curves. Monte Carlo simulations were applied to quantify the probability of bridge failure given by different lahar depths. Fragility curves of bridges were parameterized by maximum likelihood estimation, using a cumulative lognormal distribution. Bridge failure model and parameterized fragility curves were successfully validated for a 95 % confidence level using data of 15 bridges that were reached by lahars in the last 50 years. Validated models confirm that decks fail mainly due to piers and/or abutment overturning, rather than sliding forces; these models also demonstrated that bridges with piers are more vulnerable to lahars. Further research is being conducted to develop an application tool to simulate the effects of expected lahars in existing bridges of a road network.

## 1 Introduction

Volcanic eruptions produce operational losses and permanent physical damage to highway infrastructure, as evidenced by historical data regarding this natural hazard. The level of damage depends on the infrastructure's exposure and vulnerability as well as the type of

volcanic event, namely: pyroclastic fall, pyroclastic flow, lava flow and lahar. Consequences related to pyroclastic fall, specifically tephra, are temporary road closures due to the lack of visibility and loss of surface friction (Nairn, 2002; Leonard et al., 2005; Wilson et al., 2012). Lava and pyroclastic flows destroy the infrastructure but, in contrast,

their probability of occurrence is low and their influence area is small (Wilson et al., 2014). This implies a lower hazard intensity and exposure and, therefore, a lower risk of lava and pyroclastic flows on the infrastructure, considering that risk is a function of the hazard, exposure and vulnerability (UNISDR, 2009). Lahars are flows of water, rock fragments and debris that descend from the slopes of volcanoes and river valleys. The highways reached

by lahars are affected physically and operationally (Smith and Fritz, 1989). Volcanic debris and sediments transported by lahars make these flows especially destructive. These flows also scour the riverbed permanently affecting the foundations of the exposed infrastructure (Vallance and Iverson, 2015; Muñoz-Salinas et al., 2007; Nairn, 2000). Wilson et al. (2014) demonstrated that critical infrastructures affected by lahars include bridges (including piers,

foundations, abutments and deck) and culverts. Blong (1984) and Wilson et al. (2014) reported that, as consequence of Mount St. Helens (USA) eruption in 1980, 300 km of highways were damaged and 48 bridges were affected. The eruption of Villarrica and Calbuco volcanoes, which occurred in Chile in 2015, collapsed four of six bridges reached by lahars.

Several authors have calibrated fragility curves for buildings and electrical transmission systems, considering the vulnerability of both to volcanic hazard (Spence et al., 2005; Spence et al., 2007; Jenkins and Spence, 2009; Zuccaro and De Gregorio, 2013). Wilson et al. (2017) developed road infrastructure fragility curves due to tephra fall, without analyzing the effect of lahars on bridges. Fragility curves are commonly integrated in

available risk modelling tools. For example, in the United States, the Federal Emergency Management Agency (FEMA) developed HAZUS-MH tool for risk management of structures and infrastructure. This GIS-based software studies three natural hazards: earthquakes, floods and hurricanes, excluding the volcanic hazard from the analysis (FEMA, 2011). Likewise, the RiskScape software developed by the National Institute of

Water and Atmospheric Research (NIWA) of New Zealand included the effects of earthquakes, tsunamis, floods, hurricanes and volcanic eruptions over assets such as buildings, roads and power lines. Nevertheless, the effects of volcanoes are only accounted for in terms of ash fall and the temporary effects on the infrastructure operation (Kaye, 2008).

From available literature and the current state-of-the-practice, it was concluded that no failure models and fragility curves have been developed to estimate bridge failure probability due to lahar flows. Therefore, the main objective of this study was to propose a simplified bridge failure model and bridge fragility curves due to lahar hazards, considering pier and abutment overturning, as well as deck sliding. Model development considers the

calibration, parameterization and validation of bridge fragility curves due to lahars, based

on a validated limit state model. Two damage states were considered in the analysis: bridge failure and non-failure.

The research starts with the characterization of the lahar process and the physical effects on bridges. A failure model is then proposed, considering the limit state of the infrastructure overturning moment and the tangential force over the deck caused by lahars, for one-span and multiple-spans bridges. An experimental design was elaborated to calibrate fragility curves based on analytical models that characterize the effect of lahars over bridges. Monte Carlo simulations were applied to estimate the failure probability considering different lahar depths. The fragility curves were parameterized using maximum likelihood estimation, considering a cumulative lognormal distribution. Proposed bridge failure model and fragility curves were empirically validated with the available historical data. Finally, resulting curves are analyzed in detail.

## 2 Characterization of lahars for the development of fragility curves

### 2.1 Physical description of lahar flows

Lahars are high-velocity flow composed by a mix of volcanic debris and water, travelling through ravines and riverbeds (Pierson et al., 2009). Lahar flows are originated by an abrupt melting of snow and/or ice caused by the heat flow derived from lavas or pyroclastic flows issued during a volcanic event, or by avalanches of non-consolidated volcanic material during intense rains or rupture of a lake or pond (Waitt, 2013). Lahars are categorized according to their sediment/water ratio into debris flows and hyper-concentrated flows (Smith and Fritz, 1989). Debris flows are highly viscous slurries of sediment and water. Debris flows are capable of transporting gravel-sized debris in suspension, and their concentration of solid particles ranges between 75 and 80 % in weight or 55 and 60 % in volume. Hyper-concentrated flows have high-suspended fine contents, predominantly due to fluid motion and properties. The solid concentrations of hyper-concentrated flows can represent up to 55 to 60% of the total weight, and 35 to 40% of the total volume (Pierson et al., 2009).

The flow of lahars is guided by gravity, so the flow is capable of impacting elements located tens of kilometers away from the crater of the volcano (Parfitt and Wilson, 2008). Furthermore, lahars can reach velocities up to 140 km/h, as observed in Mount St. Helens in the United States in 1980 (Pierson, 1985). The velocity and composition of lahars make them highly destructive.

According to Vallance and Iverson (2015) and Bono (2014), the most important processes of a lahar are the erosion of the steep slopes and the scouring of beds of fluvial terraces. Even more significant is the erosion observed in steeper river valleys with weaker beds. Watery sediment floods are more erosive than sediment-rich flows, where scour of the riverbed drags massive material blocks (presenting diameters over 10 m) and vegetation. In

this context, most of the bridges affected by lahars are located in valleys in volcanic areas. The erosion and the associated loads of high velocity lahars, and the impact of debris travelling with them, may cause the collapse or permanent deterioration of bridges (Nairn, 2002). This explains, in part, the high vulnerability of bridges to lahar flows.

Relevant drivers of the destructive potential of a lahar affecting a bridge are the bed material, the slope, the season in which the lahar occurs, the existence of a glacier, rainfall and the prevailing temperatures during winter. The destructive potential of a lahar increases when the eruption occurs at the end of the winter, since in this season there is more accumulated snow compacted in layers, and more volume of ice melting. This condition is
enhanced if winter temperatures are low, because greater volumes of ice and snow melting in shorter lapses of time may increase the lahars' intensity (Moreno, 2015).

**2.2 Bridge fragility curves for lahar risk modelling**

In order to incorporate the uncertainty of the characteristics of lahar flows and the bridge engineering design ($X$), the use of fragility curves to quantify the probability of bridge
failure due to lahars is proposed. Fragility curves express the probability that a system exceeds different damage states ($ds_i$) as a function of the hazard intensity ($IM$) (See Eq. 1). The fragility curves allow quantifying the failure probability of a system due to an event of a specific intensity (Rossetto et al., 2013), representing the systems' vulnerability to a natural hazard.

$P(DS \geq ds_i | IM)$ ,                                                                                      (1)

Schulz et al. (2010) define four approaches for developing a system's fragility curves. First, there is the empirical approach, which is based on historical data and/or experiments. Fragility curves can be based on experts' opinions as well. Fragility curves can also be developed using an analytical approach through models that characterize the limit state of
the element, based on probabilistic and deterministic variables defining the system. Finally, a hybrid method, which combines two or more of the recently described approaches, can be used.

Since there are no existing models addressing lahar risk on bridges, a challenge for the development of bridge fragility curves consists in defining a unified lahar hazard intensity
($IM$). In general, the flow depth is a measure of hazard intensity of natural events that involve liquid flows. In the flood module of the HAZUS-MH software, the Federal Emergency Management Agency developed fragility curves using the flow depth to quantify the hazard intensity (FEMA, 2011). Tsubaki et al. (2016) use the same variable (flow depth) for measuring the flood intensity when developing embankment fragility
curves. Wilson et al. (2014) propose the flow depth as one of the potential intensity measures for developing fragility curves related to lahar flows as well. In this paper the lahar depth was proposed as lahar hazard intensity, considering that this variable is

correlated to other lahar flow characteristics, such as velocity and scour demand (Arneson et al., 2012).

## 3 Proposed failure model for infrastructure overturning and deck sliding due to lahars

### 3.1 Conceptual model

In order to model bridge fragility due to lahars, the analytical approach is used based on reliability principles. The assessment of the bridge reliability can be considered a supply and demand problem associated with a bridge-lahar system defined by its basic variables ($X$). The supply function ($S(X)$) of the bridge corresponds to its capacity to resist the loads

of the lahar. It is directly related to the design of the structural element. The demand function ($D(X)$) represents the load applied by the lahar on the bridge. The limit state function ($g(X)$) of the bridge-lahar system is given by the difference between the supply and demand functions. If g($X$) is lower than zero, the lahar loads on the structure are greater than the bridge capacity and hence, the bridge will fail.

With the purpose of conceptualizing the loads applied by the lahar flow on the bridge components, a bridge-lahar model was developed, which is shown in the free-body diagram in Fig. 1. It shows the generic cross section of a bridge, and the main physical loads applied by the lahar on the bridge. The cross section of the bridge in Fig. 1 is composed by the infrastructure (foundation and pier/abutment) and the superstructure (deck and beams). The

proposed failure model can be adapted to different bridge design criteria. In this paper, the Chilean design standards are considered for the fragility curves calibration. Thus, the proposed model assumes that the foundation has no piles. This assumption is based on the fact that 88 % of the bridges exposed to the volcanic hazard from the Villarrica and Calbuco volcanoes do not have piles (Moreno, 1999; Moreno, 2000). Additionally, it

assumes a simple support of the superstructure on the piers and abutments.

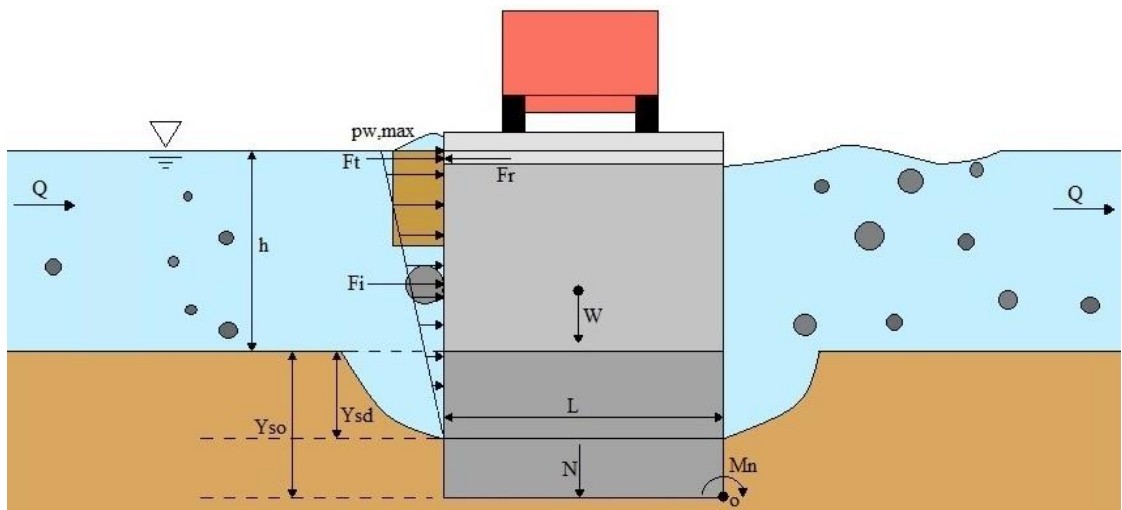

**Figure 1:** Free-body diagram of bridge resisting and demanding forces and moments in the presence of a lahar.

Fig. 1 shows a lahar with depth $h$ acting on a bridge of width $L$. Each pier or abutment of the bridge has a weight $W$. The foundation of the bridge's infrastructure was designed with a depth $Y_{so}$ that represents the supply or capacity of the bridge to resist scour. The foundation transfers loads to the ground, considering a trapezoidal distributed load model. The modelled lahar generates a hydrodynamic pressure $p_w$, which acts perpendicular to the bridge. This pressure produces a resulting hydrodynamic tangential force $F_{wi}$ on the piers and abutments, and a force $F_{ws}$ on the bridge superstructure. The deck of the bridge resists the sliding with a friction force $F_r$. The lahar also generates a scour demand $Y_{sd}$ on the bed, around the foundation. Furthermore, the debris, transported by the lahar colliding with the bridge, impacts the structure with a force $F_i$. All these forces produce a net resulting moment $M_n$ on the lower right vortex of the foundation. The net moment $M_n$ is equal to the difference between the overturning moment $M_v$ generated by hydrodynamic forces $F_{wi}$, debris impact $F_i$ and the resistant moment produced by the weight $W$ of the bridge.

### 3.2 Bridge failure mechanisms due to lahars

The hydrodynamic pressure of the lahar flow ($p_w$) and the impact force of the debris ($F_i$) can cause the overturning of bridge piers and abutments. This is further enhanced by the scour that these flows generate around the foundations. The hydrodynamic pressure of the lahars, together with the potential impact of debris, can cause deck sliding.

With the aim of analyzing the effects of lahars on bridges, failure mechanisms associated with three bridge components are defined: pier overturning, abutment overturning and sliding of the bridge superstructure. In addition to these failure mechanisms, the access embankment of the bridge may collapse. However, this component is not included in the modelling due to its lower replacement cost in relation to other bridge components. All these failure mechanisms are consistent with the postulates of Wilson et al. (2014) and the

records of the lahar effects as a result of the eruptions of the Villarrica volcano and the Calbuco volcano in 2015 (MOP, 2015a; MOP, 2015b). Images in Fig. 2 (a) and (b) show the Río Blanco Bridge (Chile) before and after a lahar flow following the eruption of Calbuco volcano in 2015. Fig. 2 shows the structural collapse of the bridge due to the overturning of the pier and subsequent sliding of the deck.

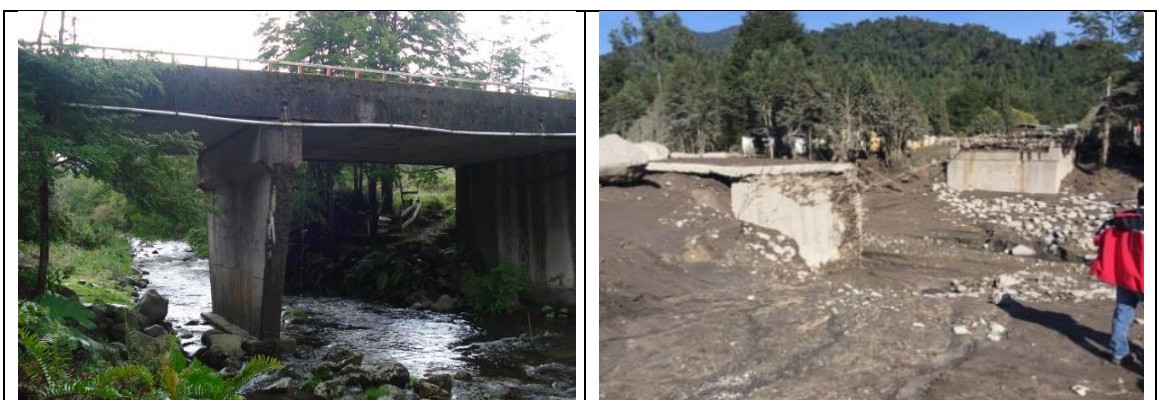

**Figure 2:** (**a**) Original Río Blanco Bridge (Chile) (MOP, 2015). (**b**) Río Blanco Bridge (Chile) after lahar flow of the Calbuco volcano eruption in 2015 (MOP, 2015).

### 3.2.1 Infrastructure overturning (piers and abutments)

Both piers and abutments are components susceptible to overturning due to lahars. These dense and fast-travelling flows generate a resulting hydrodynamic force ($F_{wi}$) on the bridge infrastructure, which entails an overturning moment ($M_{wi}$). In addition, the impact force ($F_i$) of the debris on piers and abutments produces the overturning moment ($M_i$). The bridge weight $W$ generates a moment ($M_r$) resisting the infrastructure overturning.

Through equilibrium of moments, considering the turning point $O$ located in the vertex of the foundation, it is possible to evaluate the stability of the bridge piers and abutments in the presence of a lahar flow of a specific intensity. The overturning of piers and abutments is produced if the overturning moment ($M_v = M_{wi} + M_i$) caused by the lahar on the component is greater than the resistant moment ($M_r$). In other words, the overturning is produced when the net moment ($M_n$) is less than zero.

A lahar can also cause the overturning of piers and abutments when the depth of the scour generated by the flow on the bed $Y_{sd}(X)$ is greater than the design scour of the infrastructure $Y_{so}(X)$.

The above allows establishing the limit state function $g_{VI}(X)$ related to the overturning of piers and abutments due to lahars. This function allows quantifying the overturning probability of the infrastructure considering the parameters ($X$) of the system and the lahar intensity $h_1$:

$$P_{VI} = P(g_{VI}(X) \leq 0) , \tag{2}$$

$$g_{VI}(X) = \min\{M_r(X) - M_v(X); Y_{so}(X) - Y_{sd}(X)\}, \tag{3}$$

This function indicates that, given a lahar with height $h_1$, the infrastructure will overturn if the overturning moment $M_v$ is greater than the resistant moment $M_r$ and/or the lahar scour demand $Y_{sd}$ is higher than the design scour of the bridge $Y_{so}$.

The scour produced by lahar flows near the foundations contributes to a greater vulnerability of these bridge components, since the lahars produce destabilization and weakening around the foundation of piers and abutments. If there is scour in the bed, the foundation of the pier or abutment will be exposed to a higher hydrodynamic pressure. This load is higher in the case of lahars, given their greater density and velocity in relation to
normal floods. A greater scour demand will imply a larger surface affected by the hydrodynamic pressure. In turn, this means a greater resulting hydrodynamic force ($F_{wi}$) and, therefore, a greater moment associated with this force ($M_{wi}$).

### 3.2.2 Deck sliding

In the case where the lahar height exceeds the bridge clearance, the lahar flow will exert a
hydrodynamic pressure on the bridge superstructure. There is also the possibility that the debris transported by the lahar flow impacts the bridge deck. This debris impact force ($F_{is}$), together with the hydrodynamic force ($F_{ws}$) can cause failure due to deck sliding. The presence of microscopic imperfections between the contact surfaces of the superstructure (beams) and the infrastructure (piers and abutments) produces a static friction force ($F_r$)
that opposes the start of the deck sliding.

Through the equilibrium of forces it can be inferred that the deck of a bridge subjected to a lahar will slide if the resulting tangential force ($F_t = F_{ws} + F_{is}$) is higher than the static friction force ($F_r$) between the infrastructure and the superstructure. It should be highlighted that this force is zero if the lahar height is lower than the bridge clearance.

This allows establishing the limit state function $g_{DS}(X)$ associated with the superstructure failure due to its potential sliding:

$$P_{DS} = P(g_{DS}(X) \leq 0), \tag{4}$$

$$g_{DS}(X) = F_r(X) - F_t(X), \tag{5}$$

The limit state function defined in Eq. (5) implies that, under attributes $X$, if the friction
force is lower than the tangential force produced by the lahar, the failure mechanism associated with sliding will be activated.

## 4 Experimental design for modelling infrastructure overturning and deck sliding due to lahars

### 4.1 Physical models to estimate limit state functions

Once the limit state functions have been analytically defined, the loads presented in the
free-body diagram have to be quantified. Therefore, physical existing models are used and
integrated.

### 4.1.1 Lahar hydraulic attributes

First, the lahar mean velocity ($v_{Lahar}$) is quantified with the Eq. (6), suggested by Chen
(1983; 1985) for a fully dynamic debris flow in a channel with an arbitrary geometric
shape. For this case, a rectangular flow is assumed. This formula incorporates the rheology
of the lahar through the consistency index ($\mu_{Lahar}$), which was quantified by Laenen and
Hansen (1988) for the case of lahars.

$$v_{Lahar} = \frac{2}{5}\left(\frac{\gamma_{Lahar}}{\mu_{Lahar}}\right)^{\frac{1}{2}} i^{1/2} \left(\frac{A_{Lahar}}{P_{Lahar}}\right)^{3/2}, \tag{6}$$

The lahar hydrodynamic pressure ($p_w$) is estimated with the AASHTO model (2012). This
model considers a triangular distribution of this pressure, taking a value of zero in the
deepest point and a maximum value in the flow surface. The hydrodynamic pressure is a
function of the specific weight of the flow, its velocity and the accumulation of debris ($C_D$).

$$p_{w,max} = C_D \frac{\gamma_{Lahar}}{g} v_{Lahar}^{2}, \tag{7}$$

### 4.1.2 Scour models

The lahar scour demand is based on the empirical equation proposed by Arneson et al.
(2012). Müller (1996) compared 22 equations proposed in the literature to estimate scour;
he used empirical data of 384 field measurements of 56 bridges. The conclusion of this
study was that the equation proposed by Arneson et al. (2012) in the Hydraulic Engineering
Circular No. 18 (HEC-18) was suitable for quantifying the magnitude of the scour.

Debris transported by the flows accumulates in the bridge piers, creating an additional
obstruction to the flow. To incorporate the debris accumulation, the scour demand on the
piers ($Y_{c-d}$) is modelled with Eq. (8) and (9) of the NCHRP (2010). The equations
proposed by the NCHRP adjust the scour model proposed by the HEC-18 to estimate the
scour generated by debris flows and lahars. The adjusted model considers a triangular or
rectangular debris accumulation ($K_E$) with height $H_d$ and width $W_d$ to estimate an effective
widening ($b_d^*$) of the pier with width $b$. It should be noted that factors $K_1, K_2$ and $K_3$ are
correction factors of the pier shape, the flow angle and the bed condition, respectively.

$$Y_{c-d} = 2h_{Lahar}K_1K_2K_3 \left(\frac{b}{h_{Lahar}}\right)^{0,65} Fr_{Lahar}^{0,65}, \tag{8}$$

$$b_d^* = \frac{K_E(H_dW_d)+(h_{Lahar}-K_EH_d)b}{h_{Lahar}}, \tag{9}$$

According to the HEC-18, the scour demand on the abutments ($Y_{e-d}$) is based on the flow depth, the flow width, the bridge length and a bed condition amplification factor ($\alpha$).

5 $$Y_{e-d} = \alpha h_{Lahar} \left(\frac{b_{Flow}}{L_{Bridge}}\right)^{6/7} - h_{Lahar} , \tag{10}$$

The scour supply is estimated with models adapted from bridge design manuals. For example, Breusers, Nicollet and Shen (1977) stipulate Eq. (11) and (12) assess the design scour of piers ($Y_{c-o}$) and abutments ($Y_{e-o}$). These equations include variables such as design height ($h_{design}$), pier width ($b$) and correction factors by flow angle, pier shape, 10 among others:

$$Y_{c-o} = 2b\left(K_SK_wK_gK_{gr}K_RK_d\right)tanh\left(\frac{h_{Design}}{b}\right) + 2.0 , \tag{11}$$

$$Y_{e-o} = \left(K_\phi K_FK_hK_\sigma K_I\right)h_{Design} + 2.0 , \tag{12}$$

### 4.1.3 Infrastructure overturning moment and deck tangential force

The overturning moment ($M_v$) produced by lahars on the bridge infrastructure is given by 15 the sum of the hydrodynamic moment ($M_{wi}$) and the debris impact moment ($M_i$). The tangential force ($F_t$) on the deck corresponds to the sum of the resulting force from the hydrodynamic pressure on the deck ($F_{ws}$) and the debris impact force ($F_{is}$). Considering the pressure model showed in Eq. (7), the hydrodynamic moment generated by the lahar on the infrastructure ($M_{wi}$) can be estimated. In the case of infrastructure, the hydrodynamic 20 moment is separated into two parts: the foundation and the column. This separation is supported by the fact that these elements have different geometry and that the pressure has a triangular distribution over the foundation and trapezoidal distribution over the column (Fig. 1).

$$M_{wi} = M_{w,found} + M_{w,column} = F_{w,found}y_{w,found} + F_{w,column}y_{w,column} , \tag{12}$$

25 The resulting hydrodynamic force exerted by the lahar on the foundation ($F_{w,found}$) and the height at which this force acts with respect to the turning axis ($y_{w,found}$) are given by Eq. (13) and Eq. (14):

$$F_{w,found} = LC_D \left(\frac{\gamma_{Lahar}}{2g}\right) v_{Lahar}^2 \left(\frac{Y_{sd}^2}{h_{Lahar}+Y_{sd}}\right), \tag{13}$$

$$y_{w,fuond} = Y_{so} - \frac{Y_{sd}}{3} , \tag{14}$$

30 The hydrodynamic force on the column ($F_{w,column}$) and its application point ($y_{w,column}$)

depend on if the height of the lahar exceeds the height of the column or not. To incorporate this, the variable $h^*$ was defined, which is given by the minimum between the lahar height ($h_{Lahar}$) and the column height ($h_{Design}$).

$$F_{w,column} = bC_D \left(\frac{\gamma_{Lahar}}{2g}\right) v_{Lahar}^2 \left(\frac{h^{*2}+2h^*Y_{sd}}{h_{Lahar}+Y_{sd}}\right), \tag{15}$$

$$y_{w,column} = Y_{so} + \frac{\left(\frac{h^*}{2}Y_{sd}+\frac{h^{*2}}{3}\right)}{\left(Y_{sd}+\frac{h^*}{2}\right)}, \tag{16}$$

In order to quantify the hydrodynamic force of the lahar on the deck ($F_{ws}$), three cases should be considered: (1) the lahar height is lower than the bridge clearance, (2) the lahar height is greater than the clearance but lower than the roadway level, (3) the lahar height is greater than the roadway level. In the model, the roadway level is given by the sum of the

10    infrastructure height ($h_{Design}$), and the superstructure thickness ($e_{Super}$).

$$F_{ws} = \begin{cases} 0 & h_{Lahar} < h_{Design} \\ L_{Bridge}C_D \left(\frac{\gamma_{Lahar}}{2g}\right) v_{Lahar}^2 \left(\frac{h_{Lahar}^2-h_{Design}^2}{h_{Lahar}+Y_{sd}}\right) & h_{Design} \leq h_{Lahar} < h_{Design} + e_{Super} \\ L_{Bridge}C_D \left(\frac{\gamma_{Lahar}}{2g}\right) v_{Lahar}^2 \left(\frac{2h_{Design}e_{Super}+e_{Super}^2}{h_{Lahar}+Y_{sd}}\right) & h_{Lahar} \geq h_{Design} + e_{Super} \end{cases}, \tag{17}$$

To quantify the impact of debris on the bridge, the model of Haehnel and Daly (2004) is used. This model assesses the impact force through a one-degree-of-freedom system assuming a rigid structure. Thus, the impact force of gravel transported by a lahar on the

15    bridge is based on the flow velocity ($v_{Lahar}$), the specific weight of the gravel ($\gamma_{Gravel}$), the gravel diameter ($D_{Gravel}$) and the contact stiffness of collision ($\hat{k}$). Debris impact force on the deck ($F_{is}$) is given by Eq. (18).

$$F_{is} = \begin{cases} 0 & h_{imp} < h_{Design} \\ v_{Lahar}\sqrt{\hat{k}\gamma_{Gravel}\frac{4}{3}\pi\left(\frac{D_{Gravel}}{2}\right)^3} & h_{Design} \leq h_{imp} < h_{Design} + e_{Super} \\ 0 & h_{imp} \geq h_{Design} + e_{Super} \end{cases}, \tag{18}$$

The moment of debris impact ($M_i$) on the infrastructure with respect to the rotation axis is

20    shown in Eq. (19). This indicates that if the impact height ($h_{imp}$) is greater than the infrastructure ($h_{Design}$), the associated moment is zero. For the impact height, a triangular distribution with the mode equal to the lahar height is assumed, considering that the debris tends to collect in the flow surface (Zevenbergen et al., 2007).

$$M_i = \begin{cases} v_{Lahar}\sqrt{\gamma_{Grava}\frac{4}{3}\pi\left(\frac{D_{Grava}}{2}\right)^3}\left(h_{imp} + Y_{so}\right) & h_{imp} \leq h_{Design} \\ 0 & h_{imp} > h_{Design} \end{cases}, \tag{19}$$

### 4.1.4 Infrastructure resistant moment and deck friction force

The infrastructure capacity to oppose overturning depends on the bridge elements' design and condition, including the bridge geometry, materials and the scours' design ($Y_{c-o}$ and $Y_{e-o}$). Thus, the lahar loads on the bridge and the scour are considered only in the demand function (overturning moment $M_v$). The resistant moment ($M_r$) of the infrastructure to lahars is given by the weight ($W$) of the pier or abutment and the elements that are supported on it. Among the elements supported by the infrastructure, the superstructure and the soil on the abutments' foundations must be considered. The weight of the piers and abutments without considering the soil and the superstructure are:

$$W_{Infra} = \gamma_{Infra} Y_{so} L^2 + \gamma_{Infra} h_{Design} bL , \tag{20}$$

The weight of the soil on the abutment foundation in the access to the bridge is given by Eq. (21).

$$W_{Soil-Abutment} = 0.5 \gamma_{Soil} h_{Design} (L^2 - bL) , \tag{21}$$

The model considers that the weight of the superstructure is distributed uniformly in all its supports ($NA$). Thus, the force exerted by the superstructure on each foundation is:

$$W_{Super} = \frac{(\gamma_{Super})(L)(L_{Bridge})(e_{Super})}{NA} , \tag{22}$$

Since the elements of the modeled bridge are symmetrical with respect to the vertical axis, the weight acts at a distance $L/2$ from the overturning point. Thus, the resistant moment of the infrastructure is given by the following expression:

$$M_r = (W_{Infra} + W_{Soil-Abutment} + W_{Super}) \frac{L}{2} , \tag{23}$$

Finally, the force that opposes the deck sliding corresponds to the friction between the superstructure and the infrastructure. This force is given by the Eq. (24):

$$F_r = \mu_s N_{Super} = \mu_{super} (\gamma_{Super})(L)(L_{Bridge})(e_{Super}) , \tag{24}$$

### 4.2 Values of the variables involved in the limit state functions

In order to quantify the independent variables of the limit state function, the first step is to define the nature of the variables, based on their degree of uncertainty. The system bridge-lahar presents random variables associated with lahar hazard, such as lahar density and debris accumulation. To quantify these variables, probability distribution functions are used, based on studies prepared by the Chilean National Geology and Mining Service (Sernageomin) (Castruccio et al., 2010; Bono, 2014) and the United States Geological Survey (Pierson et al., 2009; Vallance and Iverson, 2015).

Furthermore, regarding variables associated with the bridges' capacity to resist lahars, random variables are also considered due to the uncertainty in the bridge design. Goodness of fit tests were undertaken to determine the probability functions and the parameters of these variables, using the information from the Chilean bridge inventory and the Highway Manual of the Ministry of Public Works (MOP, 2016). Table 1 summarizes the values of the variables involved in the limit state functions.

**Table 1:** Variables involved in the limit state functions.

| Variable | Name | Unit | Deterministic Value/ Probabilistic Distribution | Value Reference |
|---|---|---|---|---|
| $h_{Lahar}$ | Lahar Height | m | Lahar Intensity | Hazard Intensity |
| $K_w$; $K_2$; $K_\phi$ | Flow Skew Factor | - | 1.0 | Bridge Inventory (MOP) |
| $K_\sigma$; $K_g$; $K_d$ | Granulometric Dispersion Factor | - | 1.0 | MOP (2016) |
| $K_{gr}$ | Pier Group Factor | - | Uniform (1.0; 1.9) | MOP (2016) |
| $K_R$ | Foundation Emergence Factor | - | Triangular (1.0; 1.06; 1.06) | MOP (2016) |
| $h_{Design}$ | Flow Design Depth | m | Lognormal (1.16; 0.53) - 1.0 | Bridge Inventory (MOP) |
| NP | Number of Lanes | - | 1 lane; 57.8 % <br> 2 lanes; 42.2 % | Bridge Inventory (MOP) |
| L | Bridge Width | m | Burr (4.5; 14.1; 4.9) | Bridge Inventory (MOP) |
| b | Column Width | m | Triangular (0.063L; 1.0L; 0.08L) | Bridge Inventory (MOP) |
| i | Bed Slope in Bridge | - | Uniform (1.0; 1.3) | Bono (2014) |
| $L_{Bridge}$ | Bridge Length | m | Lognormal (0.78; 2.79) | Bridge Inventory (MOP) |
| $K_1$ | Pier Shape Factor | - | Triangular (0.65; 1.2; 1.1) | Bridge Inventory (MOP) |
| $K_3$ | Bed Condition Factor | - | 1,1 | MOP (2016) |
| $K_E$ | Debris Accumulation Factor | - | Uniform Discrete (0.21; 0.79) | Zavenbergen et al. (2007) |
| $W_d$ / b | Debris Width/Pier Width Ratio | - | Normal (15.1; 8.2) | Zavenbergen et al. (2007) |
| $b_F$ / $L_B$ | Lahar Width/Bridge Length Ratio | - | Uniform (1.22; 1.83) | Self-prepared with historical data |
| $\mu_{Lahar}$ | Lahar Consistency Index | kg/m | Uniform (5; 2,260) | Laenen and Hansen (1988) |
| $K_F$ | Abutment Shape Factor | - | Triangular (0.3; 1.0; 0.75) | Bridge Inventory (MOP) |
| $K_I$ | Flow Intensity Factor | - | 1.0 | MOP (2016) |
| $C_D$ | Drag Coefficient | - | 1.4 | AASHTO (2012) |
| $\gamma_{Lahar}$ | Lahar Specific Weight | N/m$^3$ | Triangular (15,598; 19,031; 19,031) | Pierson et al. (2009) |
| $\gamma_{Gravel}$ | Gravel Specific Weight | N/m$^3$ | 24,525 | Vallance and Iverson (2015) |
| $D_{Gravel}$ | Gravel Diameter | mm | Triangular (0.031; 32.0; 2.0) | Castruccio et al. (2010) |
| k | Effective Contact Stiffness | MN/m | 14.0 | Haehnel and Daly (2004); AASHTO (2012) |
| $\gamma_{Infra}$ | Infrastructure Specific Weight | N/m$^3$ | Concrete (24,500; 61.6 %) <br> Wood (7,450; 35.8 %) <br> Steel (7,450; 2.6 %) | Bridge Inventory (MOP); Cobb (2008) |
| $\gamma_{Super}$ | Superstructure Specific Weight | N/m$^3$ | Concrete (24,500; 45.7 %) <br> Wood (7,450; 53.8 %) <br> Steel (7,450; 0.5 %) | Bridge Inventory (MOP); Cobb (2008) |
| $\gamma_{Soil}$ | Soil on Abutment Specific Weight | N/m$^3$ | Uniform (12,250; 19,600) | MOP (2016) |
| NA | Number of Deck Supports | - | 2 supports; $L_{Bridge} \leq 19.05$ m <br> 3 supports; 19.05 m $< L_{Bridge} \leq 32.10$ m <br> 4 supports; $L_{Bridge} > 32.10$ m | Bridge Inventory (MOP) |
| $\mu_{super}$ | Static Friction Infra-Super | - | Concrete-Concrete (0.50; 44.9 %) <br> Concrete-Wood (0.48; 17.1 %) <br> Concrete-Steel (0.70; 0.4 %) <br> Wood-Wood (0.35; 35.0 %) <br> Wood-Steel (0.40; 2.6 %) <br> Steel-Steel (0.80; 0.0 %) | Bridge Inventory (MOP); Cobb (2008) |
| $h_{imp}$ | Gravel Impact Height | m | Triangular (0; $h_{Lahar}$; $h_{Lahar}$) | Assumption |

## 5 Calibration and parameterization of bridge fragility curves due to lahars

### 5.1 Monte Carlo simulations for fragility curves calibration

Reliability analysis comprises analytical solution methods and numerical solution methods. Analytical solution methods feature the first-order second-moment (FOSM) method, the first-order reliability method (FORM) and the second-order reliability method (SORM). Numerical solution methods include the Monte Carlo simulation (MCS) and the response surface method (RSM). The MCS method is used to develop bridge fragility curves due to lahars. The choice of the MCS as the solution method is based on the limitations of the analytical solution methods with regard to the probability distributions of the random variables (Schultz et al., 2010). MCS allows incorporating the uncertainty of the characteristics of lahars and the structure in the quantification of the bridge failure probability, without the mentioned limitation.

With the limit state functions and variables already defined, the Monte Carlo simulations can be performed. Therefore, a fixed intensity lahar $h_1$ is considered. The probability distributions of the system's random variables imply the obtainment of different values of limit state functions $g(X)$. If this function is less than zero in a specific simulation, it means that in this simulation the bridge fails due to a lahar with intensity $h_1$. The bridge failure probability due to a lahar of intensity $h_1$ is equal to the sum of the number of simulations where function $g(X)$ is negative, divided by the number of total simulations with this intensity ($NS$) (Vorogushyn et al., 2009).

$$P_{Failure} = P(g(X) < 0|\ H = h_1) = \frac{\sum_{i=1}^{NS} k_i}{NS}\ , \tag{25}$$

$$k_i = \begin{cases} 1 & si\ g_i(X) < 0 \\ 0 & si\ g_i(X) \geq 0 \end{cases}, \tag{26}$$

Simulations with fixed intensity $h_1$ allow quantifying the failure probability of the fragility curve at the abscissa $h_1$. This experiment is carried out repeatedly for several intensity levels, to obtain the complete fragility curve for each failure mechanism identified. Specifically, 10,000 simulations were performed for each intensity level. The failure probability is quantified for lahar heights discretized every 0.25 m.

### 5.2 Calibrated bridge fragility curves due to lahars

#### 5.2.1 Fragility curves by bridge failure mechanism

Once the supply and demand functions of the failure mechanisms are defined, together with their variables, simulations are run for a fixed lahar height level $h_1$. The percentage of

simulations where function $g_{VI}(X)$ is less than zero is equivalent to the overturning probability of the infrastructure in the presence of a lahar of $h_1$. After doing this for different lahar height levels, the overturning fragility curves of the piers and abutments are obtained. The same experiment was performed for the function $g_{DS}(X)$ to calibrate the deck sliding fragility curve. Figure 3 shows the fragility curves by bridge failure mechanism.

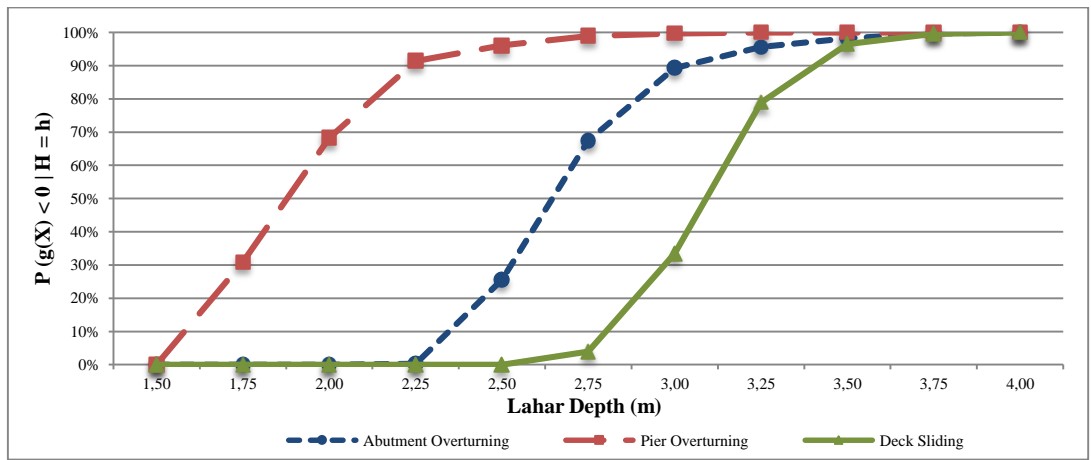

**Figure 3:** Fragility curves for bridge infrastructure overturning and deck sliding due to lahars.

The analysis of infrastructure overturning fragility curves allows us to conclude that, when impacted by lahar flows, piers are more susceptible to overturn than the abutments. Given any intensity level of the hazard, piers have a greater probability of overturning than abutments. The functional shape of the overturning fragility curves shows that, regarding the abutments; the maximum failure probability increase is achieved when the intensity grows from 2.5 to 2.75 m, where the failure probability increases 41.8 percentage points. In the case of piers, the maximum growth of the probability of failure is reached between 1.75 and 2.0 m; increasing the overturning probability by 37.4 percentage points.

When analyzing the deck sliding fragility curve, the deck failure probability is zero if the lahar intensity is less or equal to 2.50 m. This is mainly due to the fact that a low-height lahar does not reach the bridge clearance and, consequently, the flow does not affect the superstructure. Nevertheless, if the intensity of the lahar exceeds this level, the failure probability increases rapidly. The growth rate of this fragility curve also has a maximum, which is reached when the lahar arrives at 3.25 m, particularly if the lahar increases from 3.0 to 3.25 m the sliding probability of the deck increases 45.5 percentage points. This is mainly due to the fact that if the lahar reaches 3.50 m, it already touches the road elevation of most bridges of the inventory.

### 5.2.2 Fragility curves by bridge categories

The previous analysis allows us to conclude that a relevant factor in a bridge failure due to a lahar is the presence of piers. Therefore, two bridge categories were defined: bridges with one span (C1) and bridges with multiple spans (C2). Category C1 corresponds to bridges with infrastructure composed only of abutments and category C2 represents bridges with one or more piers.

To obtain the fragility curves for these two bridge categories, each simulation considered that the failure of the bridge occurs when at least one of its components fails. For example, a bridge of category C1 fails when the abutment overturns and/or when the deck slides. A category C2 bridge fails when the pier or abutment overturns and/or the deck slides. Figure 4 shows the fragility curves for both bridge categories, in addition to the failure probability of each component in a histogram.

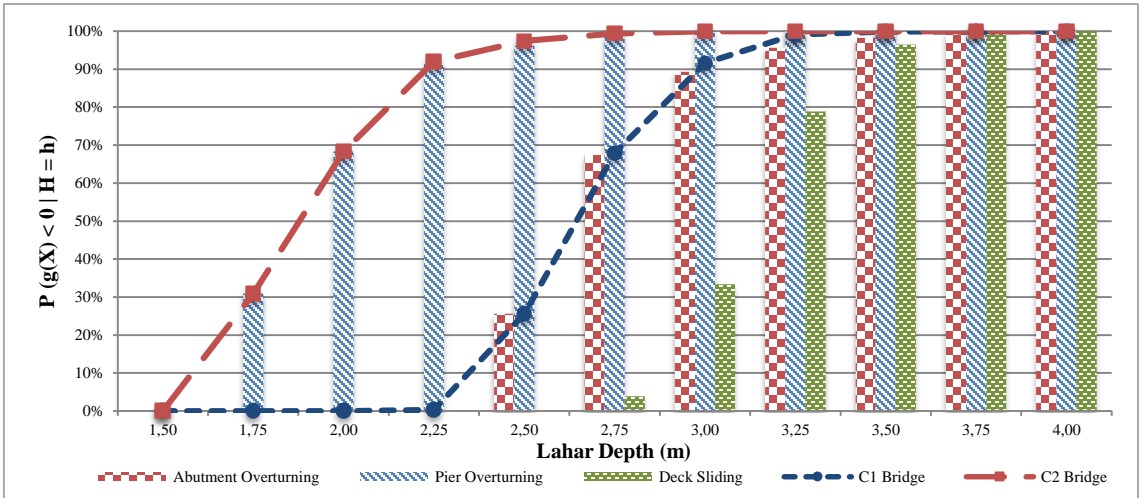

**Figure 4:** Fragility curves for one-span bridges (C1) and multiple-spans bridges (C2) due to lahars.

Fig. 4 allows us to conclude that bridges with one span (C1) are stronger than bridges with two or more spans (C2) in the presence of lahar flows. The reason is that piers are more susceptible to overturn than abutments. The failure of bridges with one span is guided by the abutments overturning, while in the bridges with multiple spans, the failure is guided by the piers overturning. The deck sliding is not a triggering factor of bridge failures due to lahars.

### 5.3 Parameterization of bridge fragility curves due to lahars

When considering risk management from a strategic point of view, the parameterization of bridge fragility curves due to lahars entails a series of advantages. It allows directly estimating the failure or collapse probability of each bridge category based on the lahar

depth. Moreover, it allows quantifying the failure probability continuously, that is, not every 25 cm of lahar.

For the parameterization of fragility curves, a cumulative lognormal distribution is considered. When assessing parameters $\mu$ and $\beta$ of the cumulative lognormal distribution reflecting the fragility curve, the bridge failure probability associated with a lahar of intensity $h_i$ can be estimated through the following equation:

$$P(g(X) < 0|H = h_i) = \Phi\left(\frac{\ln(h_i)-\mu}{\beta}\right),$$  (27)

The method of maximum likelihood estimation (MLE) was used for fragility curves parameterization. This tool allows determining the distribution parameters that maximize the occurrence probability of data obtained in the Monte Carlo simulations. In this case, the objective of the MLE is to determine the value of the bridge failure probability ($p_i$) due to a lahar of intensity $h_i$ that maximizes the probability of obtaining the pairs ($n_i$, $N_i$) associated to the simulations of all lahar intensity levels $h_i$. This is obtained by maximizing the likelihood function, which is equal to the product of the binomial probabilities for each height level $h_i$.

$$Likelihood = \prod_{i=0}^{4,0} P(n_i \text{ in } N_i \text{ collapse}|H = h_i) = \prod_{i=0}^{4,0} \binom{N_i}{n_i} p_i^{n_i}(1 - p_i)^{N_i-n_i},$$  (28)

Considering a fragility curve with cumulative lognormal distribution, $p_i$ is replaced by the cumulative lognormal function, and parameters $\mu$ and $\beta$ are estimated. In this case, it is best to maximize the likelihood logarithm instead of the likelihood function. Thus, parameters of the cumulative lognormal distribution are obtained through the following expression proposed by Lallemant et al. (2015):

$$\hat{\mu}, \hat{\beta} = argmax_{\mu,\beta} \sum_{i=0}^{4,0}\left[n_i ln\left(\Phi\left(\frac{\ln(h_i)-\mu}{\beta}\right)\right) + (N_i - n_i)ln\left(1 - \Phi\left(\frac{\ln(h_i)-\mu}{\beta}\right)\right)\right],$$  (29)

Parameters $\mu$ and $\beta$ were obtained by iterating their values and finding the combination that maximizes Eq. (29). The process was carried out for bridges with one span (C1) and bridges with multiple spans (C2). For bridges without piers (C1), the result was that the likelihood function is maximized with $\mu$ equal to 0.98 and $\beta$ equal to 0.08. In this manner, we conclude that the failure height of bridges with one span (C1) due to lahars can be modeled with a cumulative lognormal distribution ($\mu = 0.98; \beta = 0.08$). Regarding the bridges with two or more spans (C2), it was concluded that its collapse height due to lahars could be represented by a cumulative lognormal distribution with $\mu$ equal to 0.63 and $\beta$ equal to 0.13. Fig. 5 shows both analytical fragility curve and parameterized fragility curve of bridges with one span (C1) and with two or more spans (C2).

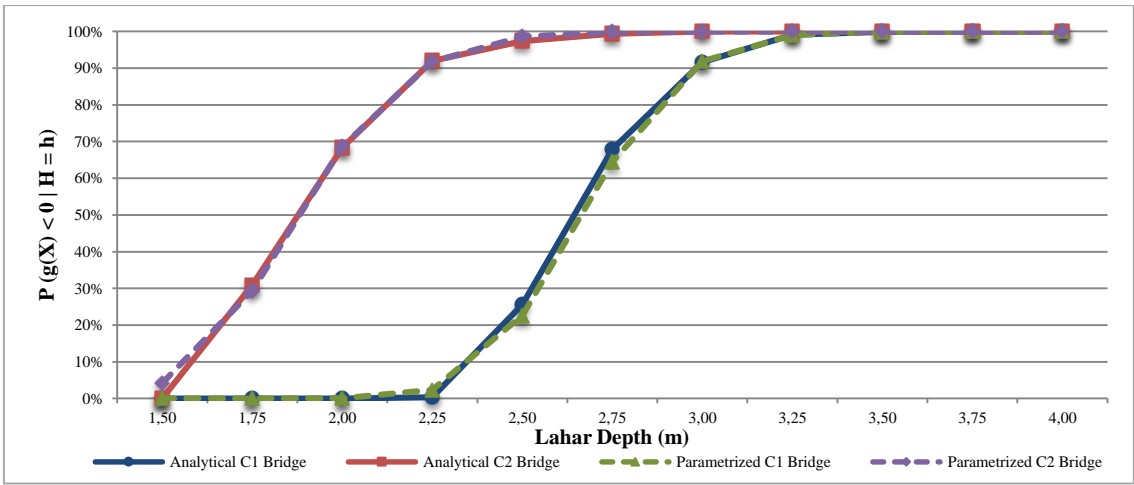

**Figure 5:** Analytical and parameterized fragility curves for one-span bridges (C1) and multiple-spans bridges (C2) due to lahars.

## 6 Validation of the bridge failure model and fragility curves due to lahars and analysis of results

### 6.1 Validation of the model of bridge failure due to lahars

The model of bridge failure due to lahars is based on physical models and expressions recommended in the literature; for example, this includes the equations given by the Highway Manual of the Chilean Ministry of Public Works (MOP, 2016) for estimating the scour supply in order to design bridges as well as the expressions of HEC-18 (Arneson et al., 2012) for quantifying the scour demand of the flows. All this requires the validation of the developed analytical failure model.

The bridge failure model is validated empirically using data from historical lahars of Chile. Considering the attributes of the historical lahars and bridges that were affected, the model quantifies the net moment ($M_n$) and net force exerted by the flow on the bridge. If the demand moment or force exceeds that of supply, the model indicates that the analyzed bridge failed due to that historical lahar. The model's result for each bridge (failure/not failure) is compared with that indicated in the damage reports. For the validation, the damage attributes and records of lahars produced during the eruptions of the Villarrica volcano in 1964, 1971 and 2015, and the Calbuco volcano in 1961 and 2015 were used. The historical information was compiled from Klohn (1963), Naranjo and Moreno (2004), Moreno, Naranjo and Clavero (2006), MOP (2015a), MOP (2015b) and Flores (2016). The results of the bridge failure model validation are shown in Table 2:

**Table 2:** Results of the bridge failure model validation.

| Nº | Bridge | Lahar Height (m) | Mn Abutment (MN-m) | Mn Pier (MN-m) | Fn Super (MN) | Analytical Damage | Empirical Damage |
|----|--------|------------------|---------------------|----------------|---------------|-------------------|------------------|
| 1 | Turbio | 3.5 | -1.98 | - | 0.62 | Failure | Failure |
| 2 | Correntoso (Villarrica) | 3.0 | -8.56 | -22.23 | -3.02 | Failure | Failure |
| 3 | Madera S/N | 5.0 | -3.71 | - | -1.09 | Failure | Failure |
| 4 | Challupén | 5.0 | -2.42 | - | -0.02 | Failure | Failure |
| 5 | El Cerdúo | 3.5 | -3.12 | - | -0.82 | Failure | Failure |
| 6 | Madera S/N 2 | 1.5 | 0.73 | 0.88 | 1.01 | No Failure | No Failure |
| 7 | Carmelito | 1.5 | 21.29 | - | 2.12 | No Failure | No Failure |
| 8 | Zanjón Seco | 1.5 | 1.99 | - | 1.81 | No Failure | No Failure |
| 9 | Seco | 1.5 | 2.43 | 0.21 | 1.36 | No Failure | No Failure |
| 10 | Tepú | 3.0 | -1.13 | -10.42 | -1.08 | Failure | Failure |
| 11 | Tronador | 3.5 | -2.04 | - | -0.18 | Failure | Failure |
| 12 | Río Blanco | 3.5 | -3.51 | - | 0.93 | Failure | Failure |
| 13 | Zapatero | 2.5 | -0.13 | - | 0.48 | Failure | Failure |
| 14 | Pescado 2 | 2.5 | 1.39 | - | 1.92 | No Failure | No Failure |
| 15 | Correntoso (Calbuco) | 2.5 | 22,16 | - | 1.49 | No Failure | No Failure |

The 15 historical cases evaluated analytically with the failure model, considering the specific inputs of the system, have the same state of damage (failure/no failure) as that reported experimentally by the agencies. It can be concluded that the bridge failure model reflects the empirical impacts of the lahars on bridges.

## 6.2 Validation of bridge fragility curves due to lahars

Once the failure model has been validated, the fragility curves must also be validated. The fragility curve validation is necessary to conclude that the probabilistic functions used in the model represent the uncertainty of the system variables. In order to validate parameterized fragility curves, the analytical bridge failure probability ($p_a$) for a lahar intensity $h_{Lahar}$ was statistically compared with the empirical failure probability ($p_e$) for the same lahar intensity. A Z-test was performed for every empirical set of points ($h_{Lahar}, p_e$) to determine whether the difference between two proportions was significant or not. Then, the following null hypothesis was proposed:

$$H_0: p_a = p_e \quad vs \quad H_a: p_a \neq p_e \ , \tag{30}$$

The empirical failure probability is estimated as the proportion of bridges reached by historical lahars with intensity $h_{Lahar}$ that were destroyed. The empirical set of points was obtained from the same information used for the bridge failure model validation (See Table 2).

Considering the null hypothesis, the test statistic $Z_{test}$ is given by the following expression:

$$Z_{test} = \frac{(\hat{p}_a - \hat{p}_e)}{\sqrt{\hat{p}(1-\hat{p})\left(\frac{1}{n_a}+\frac{1}{n_e}\right)}} \sim Normal(0,1) \ , \tag{31}$$

$$\hat{p} = \frac{x_a - x_e}{n_a + n_e}, \tag{32}$$

Where $n_a$ is the number of bridges evaluated analytically with a lahar with intensity $h_{Lahar}$ (10,000 simulations), $x_a$ the number of simulations in which the bridge fails considering an intensity $h_{Lahar}$ in the analytical model; where $n_e$ is the number of bridges that were reached empirically by lahars with intensity $h_{Lahar}$ and, $x_e$ the number of bridges that were destroyed empirically by lahars with intensity $h_{Lahar}$. The data and results of the test statistic $Z_{test}$ obtained for each hypothesis test associated with each point are shown in Table 3 and Table 4.

**Table 3:** Validation data for fragility curves of one-span bridges (C1).

| h (m) | One-span bridges (C1) | | | | | | |
|---|---|---|---|---|---|---|---|
| | $n_a$ | $x_a$ | $p_a$ | $n_e$ | $x_e$ | $p_e$ | $Z_{test}$ |
| 1.50 | 10,000 | 0 | 0.0 % | 2 | 0 | 0 | 0.00 |
| 2.50 | 10,000 | 2,265 | 22.7 % | 3 | 1 | 33.3 % | -0.44 |
| 3.50 | 10,000 | 9,993 | 99.9 % | 4 | 4 | 100.0 % | -0.03 |
| 5.00 | 10,000 | 10,000 | 100.0 % | 2 | 2 | 100.0 % | 0.00 |

**Table 4:** Validation data for fragility curves of multiple-spans bridges (C2).

| h (m) | Multiple-spans bridges (C2) | | | | | | |
|---|---|---|---|---|---|---|---|
| | $n_a$ | $x_a$ | $p_a$ | $n_e$ | $x_e$ | $p_e$ | $Z_{test}$ |
| 1.50 | 10,000 | 421 | 4.21 % | 2 | 0 | 0 | 0.30 |
| 3.00 | 10,000 | 9,938 | 99.9 % | 2 | 2 | 100.0 % | -0.01 |

Once the test statistic $Z_{test}$ of every hypothesis test associated with each point is calculated, it is compared with a significance level $\alpha$ for validation. For the fragility curve validation, a significance level of 5% was considered. The critical value ($Z_{critical}$) of ±1.96 delimits the region of acceptance and rejection of the null hypothesis. If the test statistic $Z_{test}$ is located in the acceptance region [-1.96; +1.96], the null hypothesis $H_0$, stating that the bridge empirical failure probability due to lahars is equal to that obtained by the parameterization ($H_0: p_a = p_e$); this should be accepted with that significance level. In this case, the $Z_{test}$ values of all the empirical points evaluated are within the acceptance region. The maximum absolute value obtained from $Z_{test}$ was 0.44, for one-span bridges reached by lahars of 2.50 m. Therefore, we conclude that it is possible to accept the null hypothesis $H_0$, which establishes that empirical bridge failure probability due to lahars is equal to that indicated by the analytical model, with a 5 % significance level.

## 6.3 Analysis of validated fragility curves and failure model

Once the bridge fragility curves due to lahars are calibrated, parameterized and validated, the main results obtained in the research are analyzed. First, it should be highlighted that the model of bridge failure due to lahars proposed was successfully validated for the 15 historical bridges and lahars evaluated. This allows inferring that the developed failure

model represents the fragility of its components in the presence of these flows. The null hypothesis $H_0$ was statistically accepted with a 5 % significance level, completing an empirical validation of the fragility curves. Thus, we can deduce that the modelling method based on the reliability theory and the Monte Carlo simulations can be used for calibrating bridge fragility curves due to lahars. Through the satisfactory validation we conclude that the existing models integrated in the limit state functions and the values of the used variables reflect the stability of the bridge due to a lahar flow. Finally, the validation of the parameterized fragility curves allows us to infer that the cumulative lognormal distribution with the parameters obtained through the MLE represent the bridges' fragility in case of lahars.

The analysis of the models and equations used in the limit state functions demonstrates that the lahar depth is the main variable in the quantification of lahar loads and bridge capacity to response to these flows. The lahar velocity, the scour demand, the hydrodynamic pressure and the height of the debris impact depend on the flow height. Thus, it is concluded that this variable can be used to represent the hazard intensity in the fragility curves associated to lahars.

Regarding the simulations of calibrated fragility curves for the overturning of piers and abutments, it is worthy to note the greater contribution of the moment associated with the hydrodynamic pressure than the debris impact. The average impact moment does not exceed 0.21 % of the hydrodynamic moment in the case of piers and 0.39 % for abutments. Moreover, it should be noted that the contribution percentage of the impact moment decreases as the lahar height increases.

Concerning the deck sliding, it is important to indicate that the net force is kept relatively constant when the lahar intensity is lower or equal to 2.5 m. This is because the tangential force of the lahar on the superstructure is null. Afterwards, when the lahar reaches the beams and decks, the average, minimum and maximum net forces obtained in the simulations start to decrease. For example, the average net force is negative when the lahar height is higher or equal than 3.25 m, where the failure probability is 78.9 %. Moreover, if the lahar intensity is higher or equal than 4.0 m, the deck has a 100 % probability of sliding, because the maximum net force obtained in the simulations is negative.

Furthermore, the results showed that the contribution of the force of the debris impact on the superstructure is lower in relation to the hydrodynamic force. In this particular case, the maximum average impact force represents 0.68 % of the hydrodynamic force. The reason is that the impact of debris on the superstructure is infrequent, since it requires the height of the impact to be higher than the height of the infrastructure, but lower than the road elevation. Nevertheless, if such impact should occur, the impact force would be high.

Regarding the fragility curves by bridge categories, the failure of bridges from category C2 is mainly due to the overturning of piers. In fact, when the lahar height is less or equal to 2.0 m, the pier is the only triggering component, because the other ones have no failure

probability. The failure probability of the abutments is greater than zero when the lahar intensity is greater or equal to 2.25 m. At that intensity level, the pier already has a failure probability of 91.4 %, which means that the influence of the abutment on the bridge failure is lower. That is why the fragility curve of C2 bridges is similar to that of the piers overturning.

Something similar occurs in one-span bridges (C1). In this case, the triggering component is the abutment, because it is more vulnerable to lahars than the deck. When the flow depth is higher than 2.25 m and lower than 2.5 m, the C1 bridges can fail only if the abutments overturn, since the sliding probability of the deck is zero. The deck sliding probability is no longer null at 2.75 m, reaching a sliding probability of just 3.9 %, compared with an abutment overturning probability of 67.4 %. Therefore, the abutment is always the main failure factor in this type of bridges.

## 7 Conclusions and recommendations

In this paper, a bridge failure model and bridge fragility curves due to lahars are proposed, considering pier and abutment overturning, as well as, deck sliding. The model development considers the calibration, parameterization and validation of bridge fragility curves due to lahars based on a limit state model. Two types of bridges were considered in the analysis: one-span and multiple-span bridges. Monte Carlo simulations were applied to estimate the failure probability given by different lahar depths. Fragility curves of bridges were parameterized by maximum likelihood estimation, using a cumulative lognormal distribution. Through the empirical validation of the failure model and the parameterized fragility curves, we concluded that the models included in the limit state functions and the proposed values to characterize lahar flows are representative of prevailing loads and bridge capacity.

The analysis of the validated fragility curves demonstrated that decks fail due to infrastructure overturning prior to sliding. The deck sliding probability ceases to null (3.9 %) when the lahar height is equal to 2.75 m. In the presence of a lahar of this intensity, the pier and abutment overturning probabilities are 98.9 % and 67.4 %, respectively. This implies that the probability that the deck fails and the infrastructure does not fail is 0.01 %, considering that these are independent events. In addition, the research concluded that bridges with multiple spans are more vulnerable to lahar flows compared to bridges with one span. The most evident difference between these bridges was obtained in the lahars of height 2.25 m. Given this intensity, bridges with one span (C1) have a 0.3 % probability of failure, while those with multiple spans (C2) have a 92.0 % probability of failure. This result was expected because when impacted by lahars, piers are more susceptible to overturn than abutments. With the developed fragility curves, agencies can determine the failure probability of bridges due to a lahar presenting a specific depth. The proposed failure model can be adapted and calibrated to bridge designs that are different than the

structures accounted for in the article. When required, the supply function considered in the model can be conditioned to local bridge design standards and adjusted accordingly.

For the application of these models, it is recommended that expected hazard scenarios, in terms of recurrence and intensity, should be first simulated. The resulting hazard intensity can then be estimated for the affected road network, in particular exposed bridges, and their failure probability can be consequently calculated. Further research is being conducted in this regard, where a computational platform is being developed for the consistent application of the developed fragility curves for the exposed networks. With this, local authorities can review their road and bridge designs and existing infrastructure in order to assess and apply mitigation strategies prior to the occurrence of a volcanic event.

## Acknowledgements

The authors thank the National Commission for Scientific and Technological Research (CONICYT), which has financed the FONDEF Project ID14I10309 Research and Development of Models to Quantify and Mitigate the Risk of Natural Hazards in the National Road Network. Likewise, we express our gratitude to the institutions that participated and contributed to this research project, especially to: the Research Center for Integrated Disaster Risk Management CONICYT/FONDAP/15110017 (CIGIDEN), the Highways Department of the Chilean Ministry of Public Works and the National Geology and Mining Service of the Chilean Ministry of Mining, the National Emergency Office of the Ministry of Interior and Public Security (ONEMI) and the Chilean Association of Concessionaires (COPSA).

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
