# Peer review of "Development of bridge failure model and fragility curves for infrastructure overturning and deck sliding due to lahars"

_Natural Hazards and Earth System Sciences, 2017_

## Referee Comment (RC1) · Anonymous Referee #1 · 8 Dec 2017

The manuscript discusses the fragility of road bridges to lahar hazard. The discussion started with the force and momentum balance of the bridge (section 2), the sensitivity analysis (sections 3 and 4), the development of fragility curves (section 5), and validation of the fragility curves.

On the other hand, the improper use of pronouns and adjectives causes difficulty in following the logic of the discussions. I recommend being managed this matter prior to discussing the details of the manuscript.

Examples of the improper pronouns and adjectives are follow Page 3, line 30, 'latter' what this means? fragility curve, fragility probability or others? Page 3, line 21, 'they'

what this means? Page 4, line 5, 'they' what this means? Page 18, line 6, 'they'. I think it is better to be replaced to 'we'.

Other minor suggestions. Page 1, line 21, insert 'simulation' following to 'Monte Carlo'. Page 4, line 13, I cannot understand why 'on the other hand' used here (both Tsubaki et al. and Wilson et al. use the flow depth as the hazard intensity measure. Page 4, line 13-15,' Moreover, the existing velocity and scour models use the flow height' doesn't make sense. Page 10, line 1, 'y' would be 'and'.

Pointed-out above are all minor points, but make difficult to follow the logic flow of the manuscript so discussing the detail of the present manuscript may cause many inessential discussion caused by misinterpretation of the authors' intentions. Thus, I suggest to revises the English language first.

---

## Referee Comment (RC2) · Anonymous Referee #2 · 9 Jan 2018

**Review of "Development of bridge failure model and fragility curves for infrastructure overturning and deck sliding due to lahars" by Dagá et al.**

January 9, 2018

**Summary**

This manuscript focuses on the development of fragility curves for bridges affected by lahars. The main damage mechanisms considered are pier/abutment overturning and deck sliding. Development of the fragility curves follows a process of developing a conceptual model and limit state functions, incorporating analytical models of scour and pressure and finally, Monte-Carlo simulation. These curves (for bridges without (C1) and with (C2) piers) are then parametrised and tested against bridge failure data.

While the developed fragility curves seem conceptually acceptable, there is little justification of critical assumptions, the validation is statistically unsound and manuscript is difficult to follow and understand.

My main concern is that the statistics used in section 6, which underpin the conclusion, do not support the authors assertions. The Z test used in this manuscript only provides

evidence that the null hypothesis (analytical curves match empirical data) cannot be rejected. It does not demonstrate the 'truth' of the null hypothesis. I refer the authors to the ASA statement on Statistical Significance and *P*-Values (Wasserstein and Lazar, 2016) and urge them to consider other, more suitable, approaches to validation of their fragility curves. Crucially, the number of samples and source of the empirical population is not described. I am unclear on the utility of extending data beyond 100% destruction (3.75 m/3.5 m) in both datasets as the probability of a Type I error (falsely rejecting the null hypothesis) with the authors data is 0. More concerning is the insufficient number of samples, from the abstract I believe the number of bridges in the empirical sample is 14, and am unsure on how the 'analytical' sample was taken. With such small samples, the probability of a Type II error (falsely accepting the null hypothesis) is very high. As the authors did not provide enough details on their statistical test, I am unable to calculate this, but am confident it is too high to make any meaningful conclusions.

For these reasons, I believe the manuscript needs extensive modification before it can be considered for publication. For this to be acceptable for publication, the authors need to fully justify their bridge failure model and support this model with critical analysis using their empirical data. As a suggestion, given the low number of empirical samples, it may be better to evaluate the performance of the model against their individual examples - this may provide a level of qualitative support for the model. To assist the authors, I have outlined my main concerns in the following sections.

**Main issues**

Language, structure and figures

As the other reviewer states, the grammar makes it difficult to follow the logic of this manuscript. English proof-reading is needed to ensure the minor issues of tense, pronouns and adjectives are addressed and do not confuse the reader. The use of 'on the other hand' (Page 3, line 9; Page 4, line 5; Page 8, line 2; Page 10, line 5; Page 16 , line 11 and more) also causes a lot of confusion. Figures 3 - 5 need to be modified (thinner line weights, different symbols, patterned lines) to ensure the graphics are easily readable in greyscale. Figure 1 is well drawn and designed, although definition of $Q, qmin, qmax$ is needed.

Introduction

Page 2, line 5: "This implies less exposure and therefore, vulnerability ...". This is wrong, risk is generally considered as a function of the hazard, exposure and vulnerability. In this example, the exposure and vulnerability are the same but the hazard is lower - resulting in lower risk. One could also argue that exposure is lowered, but this will not lower the vulnerability.

Page 2, paragraph 3: "From available literature..." not much literature has been explicitly surveyed here - only examples of risk management software. The Wilson (2014) review is quite extensive, but the manuscript would benefit from a broader review of available literature on bridge fragility functions.

Proposed failure model for infrastructure overturning and deck sliding and experimental design

The conceptual model for bridge failure discussed and seems reasonable, but there has been no critical analysis or justification of assumptions used. In particular:

- Page 5, line 1: The foundation has no piles. Is this justified by bridge designs (especially in your study area)? It may be a valid assumption, but the authors need to justify this with data (i.e. in the bridges used in subsequent sections, did any have piles?).

- Page 6, line 15: So you are not explicitly modelling the effect of scour on the resisting moment? Destabilisation from erosion (mentioned on Page 7, lines 1-7) would surely have a large role on changing the location or size of the moment. How is this accounted for?

- Page 7, line 25: The estimation of velocity Mannings formula is based on the assumptions of a one-dimensional, steady state flow, which is unlikely around bridges. Also, how was the effect of rheology on the flow accounted for? The velocity (and height) will depend on the rheology of the flow, this should probably be accounted for in the Monte-Carlo simulations.

- Page 7, line 27-29: How valid is a 'clear fluid' scour model for lahars? Is this model used? The grammar is unclear on page 8 (On the other hand), but if it isn't used - why is it mentioned in such detail?

- Page 8, how is the bending moment calculated? Where is the impact force located? Debris tend to collect on the surface of the flow, increasing the moment - the magnitude of this effect may be important (particularly for deck sliding).

[Figure]

In Table 1, the variables of $\gamma_{Gravel}, D_{Gravel}, h_{imp}, e_{Super}$ are not mentioned in the manuscript. How are they used in the Monte-Carlo simulations?

Validation

I refer the authors to my initial comments on the manuscript.

In equations 19 and 20, the important parameters $n_a, n_e, x_a, x_e$ are not defined or fully explained. Although $n_e$ might be assumed to be 14, what is the value of $n_a$?

On page 15, line 15 and on: At these p/Z-values, the null hypothesis is not rejected. However, this does not establish that empirical and analytical proportions are the same due to the low sample size. The significance has not been fully tested, as you have not established the statistical power of the samples.

---

## Author Response (AR1)

**REPLY TO REFEREES AND GUIDE TO THE REVISION OF THE PAPER**

**Natural Hazards and Earth System Sciences**

**Title:** Development of Bridge Failure Model and Fragility Curves for Infrastructure Overturning and Deck Sliding due to Lahars

**Authors:** Joaquín Dagá, Alondra Chamorro, Hernán de Solminihac, Tomás Echaveguren

**MS Nº:** nhess-2017-330

**Anonymous Referee #1**

**Point 1a:** **The improper use of pronouns and adjectives causes difficulty in following the logic of the discussions. I recommend being managed this matter prior to discussing the details of the manuscript. Examples of the improper pronouns and adjectives are follow Page 3, line 30, 'latter' what this means? fragility curve, fragility probability or others? Page 3, line 21, 'they' what this means? Page 4, line 5, 'they' what this means? Page 18, line 6, 'they'. I think it is better to be replaced to 'we'.**

We appreciate the comments of Referee #1 and we realize that several pronouns and adjectives were not used properly. The paper was edited and grammar was improved.

On page 3, line 21, the use of the term 'they' refers to 'debris flows'. The text was corrected as follows:

*"Debris flows are capable of transporting gravel-sized debris in suspension, and their concentration of solid particles ranges between 75 and 80 % in weight or 55 and 60 % in volume."*

On page 3, line 30, the term 'latter' refers to fragility curves; therefore, the word 'latter' was replaced by 'fragility curves'. The corrected text reads as follows:

*"In order to incorporate the uncertainty of the characteristics of lahar flows and the bridge engineering design (X), the use of fragility curves to quantify the probability of bridge failure due to lahars is proposed. Fragility curves express the probability that a system exceeds different damage states ($ds_i$) as a function of the hazard intensity (IM) (See Eq. 1)."*

On page 4, line 5, the word 'they' refers to 'fragility curves'. The text was adjusted as follows:

*"Fragility curves can also be developed using an analytical approach through models that characterize the limit state of the element, based on probabilistic and deterministic variables defining the system."*

Section 'Acknowledgements' (Page 18, line 6) was rewritten in first person. The corrected text reads as follows:

*"Likewise, we express our gratitude to the institutions that participated and contributed to this research project, especially to: […]."*

**Point 1b:** **Other minor suggestions. Page 1, line 21, insert 'simulation' following to 'Monte Carlo'.**

The term 'simulations' was included following to 'Monte Carlo' as suggested. The new text reads as follows:

*"Monte Carlo simulations were applied to quantify the probability of bridge failure given by different lahar depths."*

**Point 1c:** **Page 4, line 13, I cannot understand why 'on the other hand' used here (both Tsubaki et al. and Wilson et al. use the flow depth as the hazard intensity measure).**

We agree with Referee #1, the term 'on the other hand' generates confusion since both Tsubaki et al. and Wilson et al. recommend using the flow depth as the intensity measure. The term 'on the other hand' was removed from the sentence. The new text reads as follows:

*"Tsubaki et al. (2016) use the same variable (flow depth) for measuring the flood intensity when developing embankment fragility curves. Wilson et al. (2014) propose the flow depth as one of the potential intensity measures for developing fragility curves related to lahar flows as well."*

**Point 1d:** **Page 4, line 13-15,' Moreover, the existing velocity and scour models use the flow height' doesn't make sense.**

Again we agree with Referee #1 and the sentence was not written properly. The purpose of the phrase was to provide a justification for the use of the flow height as a measure of lahar intensity. The original sentence was modified as follows:

*"In this paper the lahar depth was proposed as lahar hazard intensity, considering that this variable is correlated to other lahar flow characteristics, such as velocity and scour demand (Arneson et al., 2012)."*

**Point 1e:** **Pointed-out above are all minor points, but make difficult to follow the logic flow of the manuscript so discussing the detail of the present manuscript may cause many inessential discussion caused by misinterpretation of the authors' intentions. Thus, I suggest to revises the English language first.**

As Referee #1 pointed-out, we agree that grammar was misleading in the initial version of the article. Language was revised and edited throughout the entire text to avoid misinterpretation.

Once again, the authors appreciate the comments made by Referee #1 and believe that the manuscript improved significantly after including the suggested adjustments.

**REPLY TO REFEREES AND GUIDE TO THE REVISION OF THE PAPER**

**Natural Hazards and Earth System Sciences**

**Title:** Development of Bridge Failure Model and Fragility Curves for Infrastructure Overturning and Deck Sliding due to Lahars

**Authors:** Joaquín Dagá, Alondra Chamorro, Hernán de Solminihac, Tomás Echeveguren

**MS Nº:** nhess-2017-330

**Anonymous Referee #2**

**Point 2a:** **As the other reviewer states, the grammar makes it difficult to follow the logic of this manuscript. English proof-reading is needed to ensure the minor issues of tense, pronouns and adjectives are addressed and do not confuse the reader. The use of 'on the other hand' (Page 3, line 9; Page 4, line 5; Page 8, line 2; Page 10, line 5; Page 16, line 11 and more) also causes a lot of confusion.**

We completely agree with Referee #2 and we realize that certain pronouns and adjectives cause confusion to the reader. We made an effort to edit and significantly improve grammar. For this, we removed from the text terms like 'on the other hand' and 'the latter'. The following table synthesizes some of the improvements included in the text:

| Page and line | Original Manuscript | New Manuscript |
|---|---|---|
| Page 3, line 9 | *"On one hand, there are debris flows, highly viscous slurries of sediment and water. They are capable of transporting gravel-sized in suspension, and their concentration of solid particles ranges between 75 and 80 % in weight or 55 and 60 % in volume. On the other hand, there are hyper-concentrated flows, which have high suspended fine contents, predominantly due to fluid motion and properties. Their solid concentrations reaches 55 to 60 % in weight and 35 to 40 % in volume (Pierson et al., 2009)."* | *"Debris flows are highly viscous slurries of sediment and water. Debris flows are capable of transporting gravel-sized debris in suspension, and their concentration of solid particles ranges between 75 and 80 % in weight or 55 and 60 % in volume. Hyper-concentrated flows have high-suspended fine contents, predominantly due to fluid motion and properties. The solid concentrations of hyper-concentrated flows can represent up to 55 to 60% of the total weight, and 35 to 40% of the total volume (Pierson et al., 2009)."* |
| Page 4, line 5 | *"On the other hand, curves can be based on experts' opinion."* | *"Fragility curves can be based on experts' opinions as well."* |
| Page 8, line 2 | *"On the other hand, the debris transported by the flows are accumulated in the bridge piers, thus creating an additional obstruction to the flow."* | *"Debris transported by the flows accumulates in the bridge piers, creating an additional obstruction to the flow."* |
| Page 10, line 5 | *"On the other hand, the numerical solution methods include the Monte Carlo simulation (MCS) and the response surface method (RSM)."* | *"Numerical solution methods include the Monte Carlo simulation (MCS) and the response surface method (RSM)."* |
| Page 16, line 11 | *"On the other hand, the analysis of the models and equations used in the limit state* | *"The analysis of the models and equations used in the limit state functions demonstrates* |

| | | |
|---|---|---|
| | *functions allow concluding that the lahar depth is the main variable in the quantification of lahar loads and bridge capacity to these flows."* | *that the lahar depth is the main variable in the quantification of lahar loads and bridge capacity to response to these flows."* |

**Point 2b: Figures 3 - 5 need to be modified (thinner line weights, different symbols, patterned lines) to ensure the graphics are easily readable in greyscale. Figure 1 is well drawn and designed, although definition of Q; qmin; qmax is needed.**

We appreciate and agree with Referee #2 that graphs needed improvement to be easily readable in greyscale. Graphs were improved considering thinner lines, patterned lines and different symbols, as suggested.

With regard to the terms Q, qmin and qmax, the authors realized that these were not required in the graphs so they were therefore eliminated.

Adjusted figures are presented below:

[Figure]

**Figure 1:** Free-body diagram of bridge resisting and demanding forces and moments in the presence of a lahar.

[Figure]

**Figure 3:** Fragility curves for bridge infrastructure overturning and deck sliding due to lahars.

[Figure]

**Figure 4:** Fragility curves for one-span bridges (C1) and multiple-spans bridges (C2) due to lahars.

[Figure]

**Figure 5:** Analytical and parameterized fragility curves for one-span bridges (C1) and multiple-spans bridges (C2) due to lahars.

**Point 2c:** **Page 2, line 5: "This implies less exposure and therefore, vulnerability...". This is wrong, risk is generally considered as a function of the hazard, exposure and vulnerability. In this example, the exposure and vulnerability are the same but the hazard is lower - resulting in lower risk. One could also argue that exposure is lowered, but this will not lower the vulnerability.**

We sincerely appreciate the comment from Referee #2 and we realize that the sentence lead to confusion. As stated by Referee #2, risk is generally considered to be a function of the hazard, exposure and vulnerability. In the original text we considered vulnerability as a function of exposure, which is also agreed by some authors (Wilson et al., 2014). The text was adjusted as suggested by the Referee #2 given that most literature agrees with the fact that exposure and vulnerability are not necessarily correlated. For this the UNISDR (2009)

definition of risk was incorporated as a reference, where risk is considered as a function of the hazard, exposure and vulnerability. The improved text reads as follows:

*"Lava and pyroclastic flows destroy the infrastructure but, in contrast, their probability of occurrence is low and their influence area is small (Wilson et al., 2014). This implies a lower hazard intensity and exposure and, therefore, a lower risk of lava and pyroclastic flows on the infrastructure, considering that risk is a function of the hazard, exposure and vulnerability (UNISDR, 2009)."*

New references:

UNISDR: UNISDR Terminology on Disaster Risk Reduction, United Nations, Geneva, Switzerland, 2009.

**Point 2d:** **Page 2, paragraph 3: "From available literature..." not much literature has been explicitly surveyed here - only examples of risk management software. The Wilson (2014) review is quite extensive, but the manuscript would benefit from a broader review of available literature on bridge fragility functions.**

We agree that limited bridge fragility models were referred to in the text, although others than Wilson et al. (2014) were reviewed by the authors initially. In order to present a broader perspective about the effects of volcanic hazard on different infrastructures as well as existing bridge fragility functions due to other hazards, authors refer to several examples of these models. The paragraph was improved as follows:

*"Several authors have calibrated fragility curves for buildings and electrical transmission systems, considering the vulnerability of both to volcanic hazard (Spence et al., 2005; Spence et al., 2007; Jenkins and Spence, 2009; Zuccaro and De Gregorio, 2013). Wilson et al. (2017) developed road infrastructure fragility curves due to tephra fall, without analyzing the effect of lahars on bridges. Fragility curves are commonly integrated in available risk modelling tools. For example, in the United States, the Federal Emergency Management Agency (FEMA) [...]."*

New references:

Jenkins, S. and Spence, R.: Vulnerability curves for buildings and agriculture, in: Technical Report D4.D for EU FP7-ENV project MIA-VITA, 2009.

Spence, R., Kelman, I., Baxter, P., Zuccaro, G. and Petrazzuoli S.: Residential building and occupant vulnerability to tephra fall, Nat. Hazard Earth Sys., 5, 477-494, 2005.

Spence, R., Kelman, I., Brown, A., Toyos, G., Purser, D. and Baxter, P.: Residential building and occupant vulnerability to pyroclastic density currents in explosive eruptions, Nat Hazard Earth Sys., 7, 219-230, 2007.

Wilson, G., Wilson, T., Deligne, N., Blake, D. and Cole, J.: Framework for developing volcanic fragility and vulnerability functions for critical infrastructure, Journal of Applied Volcanology, 6, 1-24, 2017.

Zuccaro, G. and De Gregorio, D.: Time and space dependency in impact damage evaluation of a sub-Plinian eruption at mount Vesuvius, Nat. Hazards, 68, 1399-1423, 2013.

**Point 2e:** **Page 5, line 1: The foundation has no piles. Is this justified by bridge designs (especially in your study area)? It may be a valid assumption, but the authors need to**

**justify this with data (i.e. in the bridges used in subsequent sections, did any have piles?).**

We completely agree with Referee #2 that this assumption, although valid, must be justified. Indeed, in this paper bridge design criteria of Chile are used. The proposed failure model can be adapted and calibrated for different bridge design standards. To justify the assumption that the modeled bridge does not have piles, the Chilean bridges exposed to the volcanic hazard were analyzed; we demonstrated that 88% of the Chilean bridges exposed to the volcanic hazard from the Villarrica and Calbuco volcanoes do not have piles. The new text reads as follows:

*"The proposed failure model can be adapted to different bridge design criteria. In this paper, the Chilean design standards are considered for the fragility curves calibration. Thus, the proposed model assumes that the foundation has no piles. This assumption is based on the fact that 88 % of the bridges exposed to the volcanic hazard from the Villarrica and Calbuco volcanoes do not have piles (Moreno, 1999; Moreno, 2000). Additionally, it assumes a simple support of the superstructure on the piers and abutments."*

New references:

Moreno, H.: Mapa de peligros del volcán Calbuco, Región de Los Lagos, Servicio Nacional de Geología y Minería, Documento de Trabajo Nº12, map scale 1:75.000, 1999.

Moreno, H.: Mapa de peligros del volcán Villarrica, Regiones de la Araucanía y de Los Lagos, Servicio Nacional de Geología y Minería, Documento de Trabajo Nº17, map scale 1:75.000, 2000.

**Point 2f: Page 6, line 15: So you are not explicitly modelling the effect of scour on the resisting moment? Destabilisation from erosion (mentioned on Page 7, lines 1-7) would surely have a large role on changing the location or size of the moment. How is this accounted for?**

We appreciate the comment of Referee #2 and we realize that this point was not well explained in the original manuscript. The capacity of the bridge to resist lahar loads depends on the design and condition of the bridge. The supply function (resistant moment) of the proposed failure model considers only the design criteria, without considering the bed condition. However, the scour generated by the lahar is considered in the demand function. The scour demanded by the lahar produces a greater hydrodynamic force and overturning moment. Therefore, the effect of scour causes an increase in the probability of bridge failure. This is explained in two parts of the new manuscript:

*"The scour produced by lahar flows near the foundations contributes to a greater vulnerability of these bridge components, since the lahars produce destabilization and weakening around the foundation of piers and abutments. If there is scour in the bed, the foundation of the pier or abutment will be exposed to a higher hydrodynamic pressure. This load is higher in the case of lahars, given their greater density and velocity in relation to normal floods. A greater scour demand will imply a larger surface affected by the hydrodynamic pressure. In turn, this means a greater resulting hydrodynamic force ($F_{wi}$) and, therefore, a greater moment associated with this force ($M_{wi}$).*

*[…]*

*The infrastructure capacity to oppose overturning depends on the bridge elements' design and condition, including the bridge geometry, materials and the scours' design ($Y_{c-o}$ and $Y_{e-o}$). Thus, the lahar loads on the bridge and the scour are considered only in the demand function (overturning moment $M_v$). The resistant moment ($M_r$) of the infrastructure to lahars is given by the weight ($W$) of the pier or abutment and the elements that are supported on it. […]"*

**Point 2g:** **Page 7, line 25: The estimation of velocity Mannings formula is based on the assumptions of a one-dimensional, steady state flow, which is unlikely around bridges. Also, how was the effect of rheology on the flow accounted for? The velocity (and height) will depend on the rheology of the flow; this should probably be accounted for in the Monte-Carlo simulations.**

We agree with Referee #2 that the Manning formula considers certain assumptions that limit its applicability for modeling lahar velocity around bridges. In order to improve the estimation of the lahar mean velocity, the Chen formula was used instead of the Manning formula. The Chen formula is recommended for fully dynamic debris flows because it incorporates the rheology of the lahar through the consistency index ($\mu_{Lahar}$). To explain this, the following paragraph was added in the paper:

*"First, the lahar mean velocity ($v_{Lahar}$) is quantified with the Eq. (6), suggested by Chen (1983; 1985) for a fully dynamic debris flow in a channel with an arbitrary geometric shape. For this case, a rectangular flow is assumed. This formula incorporates the rheology of the lahar through the consistency index ($\mu_{Lahar}$), which was quantified by Laenen and Hansen (1988) for the case of lahars.*

$$v_{Lahar} = \frac{2}{5}\left(\frac{\gamma_{Lahar}}{\mu_{Lahar}}\right)^{\frac{1}{2}} i^{1/2} \left(\frac{A_{Lahar}}{P_{Lahar}}\right)^{3/2}, \qquad\qquad (6)"$$

The value of the consistency index ($\mu_{Lahar}$) is also shown in Table 1. The use of the Chen formula for the estimation of lahar velocity generated minor changes in the bridge fragility curves. These changes were incorporated into the new manuscript.

New references:

Chen, C.: On frontier between rheology and mudflow mechanics, in: Proceedings of the Conference on Frontiers in Hydraulic Engineering, ASCE/M.I.T., Cambridge, MA, August 9-12, 1983, 113-118, 1983.

Chen, C.: Hydraulic concepts in debris flow simulation, in: Proceeding specialty conference delineation of landslides, flash flood, and debris flow hazards in Utah, Utah State University, Logan, Utah, 236-259, 1985.

**Point 2h:** **Page 7, line 27-29: How valid is a 'clear fluid' scour model for lahars? Is this model used? The grammar is unclear on page 8 (On the other hand), but if it isn't used - why is it mentioned in such detail?**

We appreciate the comment of Referee #2 and we understand his/her question regarding the applicability of the mentioned scour models for the case of lahars. Originally, the scour

models proposed by HEC-18 were valid for 'clear fluids.' The NCHRP adjusted the scour model proposed by the HEC-18 empirically to estimate de scour generated by debris flows and lahars. To explain this, the following paragraph was added to the paper:

*"Debris transported by the flows accumulates in the bridge piers, creating an additional obstruction to the flow. To incorporate the debris accumulation, the scour demand on the piers ($Y_{c-d}$) is modelled with Eq. (8) and (9) of the NCHRP (2010). The equations proposed by the NCHRP adjust the scour model proposed by the HEC-18 to estimate the scour generated by debris flows and lahars."*

**Point 2i:** **Page 8, how is the bending moment calculated? Where is the impact force located? Debris tend to collect on the surface of the flow, increasing the moment - the magnitude of this effect may be important (particularly for deck sliding).**

We agree with Referee #2 that lahar demand and bridge supply models are not explained in detail, thus causing confusion. In order to clearly describe the equations used in the supply and demand functions, four subsections were incorporated into section '4.1 Physical models to estimate limit state functions.' In the first subsection, velocity and hydrodynamic pressure models for lahar are detailed. In the second subsection, scour models are presented. The third subsection explains the methodology to quantify the demand function; this includes the infrastructure overturning moment and the deck tangential force. In the fourth subsection, supply functions are explained in detail.

Regarding the debris impact height ($h_{imp}$), we made a mistake in describing the probability distribution of this variable. In the Monte Carlo simulations we considered a triangular distribution with mode equal to the lahar height ($h_{Lahar}$) for the impact height instead of a uniform one.

In the following paragraphs of the subsection '4.1.3 Infrastructure overturning moment and deck tangential force,' the methodology for quantifying the overturning moment and debris impact height is explained:

*"The overturning moment ($M_v$) produced by lahars on the bridge infrastructure is given by the sum of the hydrodynamic moment ($M_{wi}$) and the debris impact moment ($M_i$). The tangential force ($F_t$) on the deck corresponds to the sum of the resulting force from the hydrodynamic pressure on the deck ($F_{ws}$) and the debris impact force ($F_{is}$). Considering the pressure model showed in Eq. (7), the hydrodynamic moment generated by the lahar on the infrastructure ($M_{wi}$) can be estimated. In the case of infrastructure, the hydrodynamic moment is separated into two parts: the foundation and the column. This separation is supported by the fact that these elements have different geometry and that the pressure has a triangular distribution over the foundation and trapezoidal distribution over the column (Fig. 1).*

$$M_{wi} = M_{w,found} + M_{w,column} = F_{w,found}y_{w,found} + F_{w,column}y_{w,column} \, , \qquad (12)$$

*The resulting hydrodynamic force exerted by the lahar on the foundation ($F_{w,found}$) and the height at which this force acts with respect to the turning axis ($y_{w,found}$) are given by Eq. (13) and Eq. (14):*

$$F_{w,found} = LC_D \left(\frac{\gamma_{Lahar}}{2g}\right) v_{Lahar}^2 \left(\frac{Y_{sd}^2}{h_{Lahar}+Y_{sd}}\right), \qquad (13)$$

$$y_{w,fuond} = Y_{so} - \frac{Y_{sd}}{3},$$ (14)

*The hydrodynamic force on the column ($F_{w,column}$) and its application point ($y_{w,column}$) depend on if the height of the lahar exceeds the height of the column or not. To incorporate this, the variable $h^*$ was defined, which is given by the minimum between the lahar height ($h_{Lahar}$) and the column height ($h_{Design}$).*

$$F_{w,column} = bC_D \left(\frac{\gamma_{Lahar}}{2g}\right) v_{Lahar}^2 \left(\frac{h^{*2} + 2h^* Y_{sd}}{h_{Lahar} + Y_{sd}}\right),$$ (15)

$$y_{w,column} = Y_{so} + \frac{\left(\frac{h^*}{2} Y_{sd} + \frac{h^{*2}}{3}\right)}{\left(Y_{sd} + \frac{h^*}{2}\right)},$$ (16)

*[...]*

*The moment of debris impact ($M_i$) on the infrastructure with respect to the rotation axis is shown in Eq. (19). This indicates that if the impact height ($h_{imp}$) is greater than the infrastructure ($h_{Design}$), the associated moment is zero. For the impact height, a triangular distribution with the mode equal to the lahar height is assumed, considering that the debris tends to collect in the flow surface (Zevenbergen et al., 2007).*

$$M_i = \begin{cases} v_{Lahar}\sqrt{\gamma_{Grava} \frac{4}{3}\pi \left(\frac{D_{Grava}}{2}\right)^3} \left(h_{imp} + Y_{so}\right) & h_{imp} \leq h_{Design} \\ 0 & h_{imp} > h_{Design} \end{cases}, \quad (19)"$$

**Point 2j: In Table 1, the variables of GammaGravel; DGravel; himp; eSuper are not mentioned in the manuscript. How are they used in the Monte-Carlo simulations?**

We appreciate the comment of Referee #2 and we realize that these variables were not well explained in the original manuscript. By improving the explanation of the supply and demand functions in section '4.1 Physical models to estimate limit state functions', the explanation of all variables was also improved considerably.

The gravel specific weight ($\gamma_{Gravel}$), the gravel diameter ($D_{Gravel}$) and the impact height ($h_{imp}$) are used to estimate the force and moment of debris impact ($F_{is}$ and $M_i$). The superstructure thickness ($e_{Super}$) is used to estimate the force exerted by the superstructure on each foundation ($W_{Super}$) and the friction between the superstructure and the infrastructure ($F_r$). In order to explain this, the following paragraphs and equations were added in section '4.1 Physical models to estimate limit state functions':

*"In order to quantify the hydrodynamic force of the lahar on the deck ($F_{ws}$), three cases should be considered: (1) the lahar height is lower than the bridge clearance, (2) the lahar height is greater than the clearance but lower than the roadway level, (3) the lahar height is greater than the roadway level. In the model, the roadway level is given by the sum of the infrastructure height ($h_{Design}$), and the superstructure thickness ($e_{Super}$).*

$$F_{ws} = \begin{cases} 0 & h_{Lahar} < h_{Design} \\ L_{Bridge}C_D \left(\frac{\gamma_{Lahar}}{2g}\right) v_{Lahar}^2 \left(\frac{h_{Lahar}^2 - h_{Design}^2}{h_{Lahar} + Y_{sd}}\right) & h_{Design} \leq h_{Lahar} < h_{Design} + e_{Super} \\ L_{Bridge}C_D \left(\frac{\gamma_{Lahar}}{2g}\right) v_{Lahar}^2 \left(\frac{2h_{Design}e_{Super} + e_{Super}^2}{h_{Lahar} + Y_{sd}}\right) & h_{Lahar} \geq h_{Design} + e_{Super} \end{cases}, \quad (17)$$

*To quantify the impact of debris on the bridge, the model of Haehnel and Daly (2004) is used. This model assesses the impact force through a one-degree-of-freedom system assuming a rigid structure. Thus, the impact force of gravel transported by a lahar on the bridge is based on the flow velocity ($v_{Lahar}$), the specific weight of the gravel ($\gamma_{Gravel}$), the gravel diameter ($D_{Gravel}$) and the contact stiffness of collision ($\hat{k}$). Debris impact force on the deck ($F_{is}$) is given by Eq. (18).*

$$F_{is} = \begin{cases} 0 & h_{imp} < h_{Design} \\ v_{Lahar}\sqrt{\hat{k}\gamma_{Gravel}\frac{4}{3}\pi\left(\frac{D_{Gravel}}{2}\right)^3} & h_{Design} \le h_{imp} < h_{Design} + e_{Super} \\ 0 & h_{imp} \ge h_{Design} + e_{Super} \end{cases} \qquad (18)$$

*The moment of debris impact ($M_i$) on the infrastructure with respect to the rotation axis is shown in Eq. (19). This indicates that if the impact height ($h_{imp}$) is greater than the infrastructure ($h_{Design}$), the associated moment is zero. For the impact height, a triangular distribution with the mode equal to the lahar height is assumed, considering that the debris tends to collect in the flow surface (Zevenbergen et al., 2007).*

$$M_i = \begin{cases} v_{Lahar}\sqrt{\gamma_{Grava}\frac{4}{3}\pi\left(\frac{D_{Grava}}{2}\right)^3}\left(h_{imp} + Y_{so}\right) & h_{imp} \le h_{Design} \\ 0 & h_{imp} > h_{Design} \end{cases} \qquad (19)$$

*[…]*

*The model considers that the weight of the superstructure is distributed uniformly in all its supports (NA). Thus, the force exerted by the superstructure on each foundation is:*

$$W_{Super} = \frac{(\gamma_{Super})(L)(L_{Bridge})(e_{Super})}{NA}, \qquad (22)$$

*[…]*

*Finally, the force that opposes the deck sliding corresponds to the friction between the superstructure and the infrastructure. This force is given by the Eq. (24):*

$$F_r = \mu_s N_{Super} = \mu_{super}(\gamma_{Super})(L)(L_{Bridge})(e_{Super}), \qquad (24)"$$

**Point 2k: In equations 19 and 20, the important parameters na; ne; xa; xe are not defined or fully explained. Although ne might be assumed to be 14, what is the value of na?**

We agree with Referee #2 that the validation parameters should be defined and explained in the manuscript. In order to improve validation description, the following paragraph was added:

*"Where $n_a$ is the number of bridges evaluated analytically with a lahar with intensity $h_{Lahar}$ (10,000 simulations), $x_a$ the number of simulations in which the bridge fails considering an intensity $h_{Lahar}$ in the analytical model; where $n_e$ is the number of bridges that were reached empirically by lahars with intensity $h_{Lahar}$ and, $x_e$ the number of bridges that were destroyed empirically by lahars with intensity $h_{Lahar}$. The data and results of the test statistic $Z_{test}$ obtained for each hypothesis test associated with each point are shown in Table 3 and Table 4."*

Additionally, Tables 2 and 3 of the original manuscript were modified to clearly show the

values of the empirical validation parameters ($n_a$, $n_e$, $p_a$, $p_e$) of the fragility curves.

**Point 2l: On page 15, line 15 and on: At these p/Z-values, the null hypothesis is not rejected. However, this does not establish that empirical and analytical proportions are the same due to the low sample size. The significance has not been fully tested, as you have not established the statistical power of the samples.**

We really appreciate the comment of Referee #2 and we realize that the validation should be improved. In the new paper, the section '6 Validation of bridge failure model and fragility curves due to lahars and analysis of results section' was separated into three subsections. The first subsection corresponds to the validation of the bridge failure model. In this first subsection, the limit state functions defined for the system are empirically validated. For this, data from historical lahars of Chile is used. Considering the attributes of the historical lahars and bridges reached, the model quantifies the net moment and net force exerted by the flow on the bridge. If the demand moment or force exceeds that of supply, the model indicates that the analyzed bridge failed due to that historical lahar. The model result for each bridge (failure/not failure) is compared with that indicated in the damage reports. The 15 historical cases evaluated analytically with the failure model, considering the specific inputs of the system, have the same state of damage (failure/no failure) as that reported experimentally by the agencies.

In the second subsection, the empirical validation of the fragility curves is presented. Through an investigation of reports of historical lahars and their damages in bridges, six empirical points of probability of failure were built ($h_{Lahar}, p_e$). These points were built from 15 reports of bridges reached by lahars in Chile. In the new manuscript, a level of significance of 5% was defined for the statistical test that compares the probability of analytical failure and the probability of empirical failure. Considering the defined significance level, $Z_{test}$ values of all the empirical points evaluated are within the acceptance region. We therefore concluded that it is possible to accept the null hypothesis $H_0$, which establishes that empirical bridge failure probability due to lahars is equal to that indicated by the analytical model, with a 5 % significance level. Additionally, an effort was made to improve the explanation of the process of statistical validation of the fragility curves.

The last paragraph of the fragility curves validation reads as follows:

*"Once the test statistic $Z_{test}$ of every hypothesis test associated with each point is calculated, it is compared with a significance level $\alpha$ for validation. For the fragility curve validation, a significance level of 5% was considered. The critical value ($Z_{critical}$) of ±1.96 delimits the region of acceptance and rejection of the null hypothesis. If the test statistic $Z_{test}$ is located in the acceptance region [-1.96; +1.96], the null hypothesis $H_0$, stating that the bridge empirical failure probability due to lahars is equal to that obtained by the parameterization ($H_0: p_a = p_e$); this should be accepted with that significance level. In this case, the $Z_{test}$ values of all the empirical points evaluated are within the acceptance region. The maximum absolute value obtained from $Z_{test}$ was 0.44, for one-span bridges reached by lahars of 2.50 m. Therefore, we conclude that it is possible to accept the null hypothesis $H_0$, which establishes that empirical bridge failure probability due to lahars is equal to that indicated by the analytical model, with a 5 % significance level."*

In the third subsection, the results obtained in the validation process of the failure model

and the fragility curves are analyzed.

Once again, the authors appreciate the comments made by Referee #2 and believe his/her suggestions and observations have greatly improved the manuscript.

---

## Referee Report (RR1)

**Review of major revision to "Development of bridge failure model and fragility curves for infrastructure overturning and deck sliding due to lahars" by Dagá et al.**

April 23, 2018

**Summary and recommendation**

This revised manuscript focuses on the development of fragility curves for *Chilean* bridges affected by lahars. The main damage mechanisms considered are pier/abutment overturning and deck sliding. Development of the fragility curves follows a process of developing a conceptual model and limit state functions, incorporating analytical models of scour and pressure and finally, Monte-Carlo simulation. These curves (for bridges without (C1) and with (C2) piers) are then parametrised and tested against bridge failure data. I believe this is a useful contribution, as while results are specifically for Chilean bridges (where the authors have data), the method and process is outlined well enough for it to be translated to other regions. Previous concerns highlighted by myself (Referee #2) and referee #1 have been addressed (see below for some smaller issues) and the language has greatly improved. The additional detail on methodology and validation, as requested, clarifies many of the previously highlighted issues. However, validation of the fragility curves (Section 6.2) is fatally flawed:

The number of empirical samples $n_e$ in each test (Tables 3 and 4)is too small to draw any statistical conclusion. From $n_a$, $x_a$, $n_e$ and $x_e$ provided, the power of each $Z$ test is around 0.05 (5%). That is, the probability of a Type II error (not rejecting the null hypothesis when it is actually false) is around 95%. Note this does not necessarily mean that the parametric curves (and failure model) are incorrect, but it indicates there is insufficient empirical data to provide a statistical validation of the model.

The lack of data is common in similar studies of fragility, and is not a problem which I expect the authors to solve. Rather, the bridge failure model should be well grounded in physical principles and critically evaluated against any sources of data. From my assessment, I believe the failure/limit state model is sufficiently comprehensive (keeping in mind the need for some level of simplicity) and the evaluation against empirical data in section 6.1 provides some

level of (qualitative) support. Therefore, I recommend the manuscript could be published, subject to the following critical changes:

- Section 6.2, and all references to the statistical validation (p. 1, lines 27-29; section 6.3 first paragraph; p. 22, line 16) should be removed completely.

- In section 6.3, the authors should additionally highlight the data-deficiency issue (possibly as a source of future work) and critically evaluate the success of the model in reference to Table 2. How representative is the empirical data of the range of conditions used to generate the analytical fragility curves? Are the main sources of force/vulnerability sufficiently explored with the empirical data and/or could qualitative 'bounds' on the reliability of these curves be determined?

**Minor/typographic issues**

- p.1 Line 29: "..that were reached..." change to "affected"

- p.2 Line 6: This implies a lower *hazard* only, not hazard intensity. Remove "intensity".

- p.3 Line 6: "...experimental design was elaborated..." not sure what this means here - possibly reword.

- p.6 Figure 1: What is $F_t$?

- p.9 Line 10: Change to "...a rectangular shape is assumed."

- p.10 Equation 14: Change to $Y_{w,found}$ (misspelt as *fuond*)

- Equations 13, 20, 21, 24 use the parameter $L$, which is bridge width. Introduce it at the first instance (Eq. 13) and I would suggest changing it to a less confusing variable (perhaps $T$ for thickness).

- p.13 Table 1: Variable $e_{super}$ is not listed on the table - what is its value?

- p.16 Line 20: This is a valid statement for your study, although I believe it may vary with type and depth of foundations (e.g. the use of piers).

- p.17 Line 29: Remove "...it was concluded that its..."

---

## Author Response (AR2)

[revised manuscript text omitted]

**REPLY TO REFEREES AND GUIDE TO THE REVISION OF THE PAPER**

**Natural Hazards and Earth System Sciences**

**Title:** Development of Bridge Failure Model and Fragility Curves for Infrastructure Overturning and Deck Sliding due to Lahars

5   **Authors:** Joaquín Dagá, Alondra Chamorro, Hernán de Solminihac, Tomás Echaveguren

**MS Nº:** nhess-2017-330

**Anonymous Referee #1**

The authors appreciate the comments made by Referee # 1. In this version of the paper, the text, figures, tables and equations were adjusted taking into account all the suggestions of the
10   referees. In addition, the writing, punctuation and English level were improved.

**Point 1.1: Title: the concept of 'bridge failure model', the difference between 'bridge failure' and 'infrastructure overturning and deck sliding', the definition of 'infrastructure' are not clear, this leads the reader to imagine what will come from the manuscript from the title difficult. My suggestion is, something like, 'Development of**
15   **failure model and fragility curves for road bridges under lahar impact' or simpler 'Development of fragility curves for road bridges under lahar impact'.**

We appreciate the suggestion of Referee #1 and we realize that the title is not completely clear to a potential reader. The 'infrastructure' concept can be confusing in the title. The adjusted title reads as follows:

20   *"Development of fragility curves for road bridges exposed to volcanic lahars"*

**Point 1.2: The ambiguity of terminologies makes difficult to follow the meaning of not only title but remaining part of the manuscript. E.g. 'infrastructure' covers a broad sense. Road network itself can be included in the concept of 'infrastructure' (as used in the first line of the abstract). 'Infrastructure' is mainly used to mean a foundation of**
25   **something in this manuscript, especially for a bridge. If so, it is better to describe 'bridge infrastructure' or 'bridge foundation'.**

We completely agree with Referee #1 that the term 'infrastructure' was used ambiguously to refer to the infrastructure in general (roads, bridges, buildings, etc.) and to the part of the bridge that supports the superstructure (piers and abutments).

30   We reviewed in detail the definitions of the bridge elements given by the LRFD Bridge Design Specifications of American Association of State Highway and Transportation Officials (AASHTO, 2012) and to refer to the piers and abutments the term 'substructure' should be used instead of 'infrastructure'. According to AASHTO (2012), substructure is the structural part of the bridge that support the horizontal span (i.e. piers and abutments). Thus,
35   the term 'infrastructure' now is used only to refer to the infrastructure in general (roads, bridges, buildings, etc.) and 'substructure' to piers and abutments. The term 'substructure' is defined in section 1:

*"To characterize bridge fragility to lahars, failure probability of primary structural elements is required, namely: substructure (i.e. piers and abutments) and deck."*

**Point 1.3: The word 'abutment' has potentially two meaning, one is the foundation of bridge pier and other is the marginal area between a river bank and the bridge. So the reader wonders what you want to say using the word 'abutment', so it is needed to clarify which do you want to mean or use another word.**

5   We agree with Referee #1, the term 'abutment' generates confusion since it has potentially two meanings. In order to clarify this, in the introduction we formally define the most vulnerable elements of bridges due to lahars (piers, abutments and deck). The LRFD Bridge Design Specifications of American Association of State Highway and Transportation Officials (AASHTO, 2012) are used for the elements definitions. The definitions of the
10  elements read as follows:

*"Piers are columns designed to be an interior support for a bridge superstructure; abutments are the end support for a bridge superstructure; and deck is the component that supports wheel loads directly and is supported by other components (AASHTO, 2012)."*

**Point 1.4: You used 'failure model' (singular) so maybe you want to define 'failure**
15  **model' as the toolbox to identify damage or non-damage for all possible types of failure mode. On the other hand, 'fragility curves' (plural) were made for each failure type. This means, the failure model was actually composed of several sub-models corresponding to each failure mode, and the sub-models were used to develop each fragility curve, if my understanding is correct. What I want to say is that it is better to**
20  **use 'failure models' instead of 'failure model'.**

Again we agree with Referee #1. Indeed, the failure model was composed of several sub-models and each sub-model was used to develop a fragility curve. Thus, it is better to use 'failure models' instead of 'failure model'. We adopted the suggestion of the referee and the terms 'failure model' are now in plural. For example, the second sentence of the abstract
25  reads as follows:

*"In this paper, bridge failure models due to lahars are proposed and, based on these, fragility curves are developed."*

**Point 1.5: Page 1, line 35, I'd like to replace 'losses' --> 'restrictions' and remove 'permanent'.**

30  We completely agree that the terms used in that sentence are not precise. The effects of volcanic eruptions on roads are operational restrictions and physical damage. The sentence was adjusted as follows:

*"Volcanic eruptions produce operational restrictions and physical damage to road infrastructure."*

35  **Point 1.6: Page 1, line 35, It is not clear how 'highway' and 'road' distinguished in the manuscript.**

We agree that the terms 'highway' and 'road' could generate confusion in the reader. To avoid this, now only the general term 'road' is used. Thus, the terms 'highway' were replaced by 'road' in the paper. The adjusted sentence reads as follows:

40  *"Volcanic eruptions produce operational restrictions and physical damage to road infrastructure."*

**Point 1.7: Page 2, line 2, 'temporary' --> 'temporal'?**

We appreciate the comment of Referee #1. The text was adjusted as follows:

*"Consequences related to pyroclastic fall, specifically tephra, are temporal road closures caused by visibility limitations and reduced friction between pavement and tires (Nairn,*

5    *2002; Leonard et al., 2005; Wilson et al., 2012)."*

**Point 1.8: Page 2, line 3, 'loss of surface friction' --> 'loss of friction between (road) pavement and tires'?**

We agree with Referee #1 that in this sentence we should specify the surfaces that lose friction due to pyroclastic fall. In order to detail this, the sentence was adjusted as follows:

10   *"Consequences related to pyroclastic fall, specifically tephra, are temporal road closures caused by visibility limitations and reduced friction between pavement and tires (Nairn, 2002; Leonard et al., 2005; Wilson et al., 2012)."*

**Point 1.9: Page 2, line 10, 'affect(ed)' is quite neutral but here you want to say some negative meaning so ''damaged' or something like this may be better to use here (not**

15   **only here but other sentences using 'affect(ed)').**

We completely agree with Referee #1. Indeed, in this sentence we want to highlight the negative effects of lahars on roads. Thus, we replaced the term 'affect(ed)' (quite neutral) by the term 'damaged' (negative). The adjusted sentence reads as follows:

*"Road infrastructures reached by lahars are damaged physically and operationally (Smith*

20   *and Fritz, 1989)."*

**Point 1.10: Page 2, line 14, The meaning of 'critical' here is not clear.**

We agree that in this sentence the term 'critical' was not clear. To avoid confusion, we decided to replace the term 'critical' by 'most exposed and vulnerable'. The adjusted text reads as follows:

25   *"Wilson et al. (2014) demonstrated that bridges and culverts are the road infrastructures elements most exposed and vulnerable to lahars."*

**Point 1.11: Page 2, line 16, put 'a' before 'consequence'.**

We appreciate again the comment of Referee #1. The corrected sentence reads as follows:

*"Blong (1984) and Wilson et al. (2014) reported that 300 km of roads were damaged and 48*

30   *bridges were affected because of Mount St. Helens (USA) eruption in 1980."*

**Point 1.12: Page 2, line 27, 'studies' --> 'covers'?**

We completely agree that in this case the term 'covers' is better than 'studies' to refer to the scope of the software HAZUS-MH. Thus, the sentence was adjusted as follows:

*"This GIS-based software covers three natural hazards: earthquakes, floods and hurricanes,*

35   *excluding the volcanic hazard from the analysis (FEMA, 2011)."*

**Point 1.13: Page 2, line 33, 'effects' --> 'interruption'**

We agree with Referee #1 that the term 'effects' should be replaced by 'interruption'. The term 'effects' was duplicated in the sentence. The adjusted text reads as follows:

*"Nevertheless, the effects of volcanoes are only accounted for in terms of ash fall and the temporary interruption of infrastructure operation (Kaye, 2008)."*

**Point 1.14: Page 2, line 36, 'and' --> 'nor'?**

We agree with Referee #1 that this sentence was not written correctly. The text was adjusted as follows:

*"From available literature and the current state-of-the-practice, it is concluded that no bridge fragility curves exposed to lahar flows have been developed."*

**Point 1.15: Page 2, line 40, It is not clear how 'calibration' and 'parameterization' distinguished.**

We sincerely appreciate the comment from Referee #1. Indeed, the terms 'calibration' and 'parameterization' could generate confusion. In order to clarify the concepts, the term 'parameterization' is described in the section 1. The adjusted text reads as follows:

*"Best-fit probability functions are finally proposed, considering cumulative lognormal distribution and their corresponding parameters from maximum likelihood analysis (parameterization)."*

**Point 1.16: Page 3, line 7, What do you mean by 'analytical models'?**

We appreciate the comment of Referee #1. The term 'analytical models' refers to models that characterize the limit state of the element (bridge), based on probabilistic and deterministic variables defining the system (lahar-bridge).

We note that this term ('analytical models') could be confusing for the reader, because this type of models is not yet defined in the text. Thus, we decided to remove it from this sentence and let the general term 'models'. Analytical models are defined later (section 2.2). The adjusted sentence reads as follows:

*"Failure models are then proposed, considering the limit state of pier and abutment overturning due to lahar demanding forces and reduced supply moment caused by scour."*

**Point 1.17: Page 3, lines 21-22, 'Debris flows are highly viscous slurries of sediment and water'. I think 'debris' is explained but not about 'flows'.**

We appreciate the comment of Referee #1. As noted in the text, Smith and Fritz (1989) categorized lahars according to their sediment/water ratio into: (1) debris flows and (2) hyper-concentrated flows. These are the two types of lahars. Both are defined between lines 21 and 27 on page 3.

*Debris flows: are highly viscous slurries of sediment and water. Debris flows are capable of transporting gravel-sized debris in suspension, and their concentration of solid particles ranges between 75 and 80 % in weight or 55 and 60 % in volume.*

*Hyper-concentrated flows: have high-suspended fine contents, predominantly due to fluid motion and properties. The solid concentrations of hyper-concentrated flows can represent up to 55 to 60% of the total weight, and 35 to 40% of the total volume.*

We hope the question of Referee #1 has been answered.

**Point 1.18: Page 3, line 28, I think you can remove 'so'.**

We agree with Referee #1 that the term 'so' can be removed. The text was adjusted as follows:

*"The flow of lahars is guided by gravity and is capable of impacting elements located tens of kilometers away from the crater of the volcano (Parfitt and Wilson, 2008)."*

**Point 1.19: Page 3, lines 36-37, 'scour of the riverbed drags massive material blocks' doesn't make sense.**

We sincerely appreciate the comment of Referee #1. The sentence was not written correctly. The text was adjusted as follows:

*"Watery sediment floods are more erosive than sediment-rich flows. The scour of the riverbed drags material blocks and vegetation."*

**Point 1.20: Equation (1): Is DS defined? What is the meaning of subscript i?**

We appreciate the comment of Referee #1. The variable 'DS' was not defined in the paper and the meaning of the subscript 'i' was not explained. A definition of both elements was added. The text was adjusted as follows:

*"Fragility curves express the probability that the damage state (DS) of a system exceeds different levels ($ds_i$; i=slight, moderate, extensive or complete), given a certain hazard intensity (IM) (See Eq. 1)."*

**Point 1.21: Page 4, line 26, 'the recently described approaches' --> 'the approaches described above'?**

Again we agree with Referee #1 that the sentence was not written properly. The corrected text reads as follows:

*"Finally, a hybrid method, which combines two or more of the approaches described above, can be used."*

**Point 1.22: Title of Section 3. I'd like to suggest to replace as '(Development of) failure model(s) for bridge pier/abutment overturning and deck sliding due to lahars'**

We appreciate the suggestion of Referee #1. The title of Section 3 was not clear, because of the confusion that the 'infrastructure' concept can generate for the reader. The corrected title reads as follows:

*"3 Development of failure models for bridge pier/abutment overturning and deck sliding due to lahars"*

**Point 1.23: Page 5, The first paragraph of section 3.1, so do you define that g(X)=D(X)-S(X)? If so, it is better to describe so using equation.**

We sincerely appreciate the comment from Referee #1. Indeed, the limit state function is defined as the difference between the supply and demand functions (i.e. g(X)=D(X)-S(X)). In order to better describe this relationship, an equation was added, as suggested by Referee #1. The adjusted text reads as follows:

*"The limit state function ($g(X)$) of the bridge-lahar system is given by the difference between the supply and demand functions ($D(X) - S(X)$). If $g(X)$ is lower than zero, the lahar loads*

*on the structure are greater than the bridge capacity and hence, the bridge will fail."*

**Point 1.24: Figure 1: It is not clear how the abutment overturning and the pier overturning were distinguished. Maybe, it is better to specify where are the deck, pier and abutment in Figure 1. Can we discuss the pier overturning using the moment Mn defined in Figure 1? Can you define Fwi, Fws, Fr in Figure 1?**

We appreciate the comment from Referee #1. Figure 1 shows a generic cross section of a pier or abutment and the deck. To clearly show where each element is in the figure, the deck, the column and the foundation are now indicated. The main forces of the system are also shown ($F_t$, $F_r$, $F_i$, etc.). Figure 1 was adjusted as follows:

[Figure]

**Figure 1:** *Free-body diagram of bridge resisting and demanding forces and moments in the presence of a lahar.*

**Point 1.25: Equations (3) and (5) Are Yso and Fr function of X? If so, how do you change Yso and Fr depending on X?**

We sincerely appreciate the comment from Referee #1. 'X' is the basic variables vector of the bridge-lahar system. The supply (overturning moment $M_r$, friction force $F_r$ and scour supply $Y_{so}$) and demand functions (overturning moment $M_v$, tangential force $F_t$ and scour demand $Y_{sd}$) depend on these basic variables (X).

For example, the scour supply ($Y_{so}$) depends on the design height, pier width and correction factors by flow angle, pier shape, among others (Eqs. 11 and 12). The friction force ($F_r$) depends on the bridge specific weight, superstructure thickness, bridge length, among others (Eq. 25).

**Point 1.26: Page 8, line 5, 'produced' --> 'caused'**

Again we agree with Referee #1. The term 'produced' should be replaced by 'caused'. The text was adjusted as follows:

*"The scour caused by lahar flows near the foundations contributes to a greater vulnerability of these bridge components, since the lahars produce destabilization and weakening around the foundation of piers and abutments."*

**Point 1.27: Title of Section 4. It is not clear what do you want to mean by 'experiment'.**

We completely agree with Referee #1 that the term "experiment' generates confusion. The title of section '4 Experimental design for modelling infrastructure overturning and deck sliding due to lahars' was replaced by '4 Proposal for modelling substructure overturning and deck sliding due to lahars'. In this section the physical models integrated in the bridge failure model are detailed and the values of the basic variables are indicated.

**Point 1.28: Page 9, line 10, 'For this case' --> 'In this study'?**

We completely agree with Referee #1 that the term 'For this case' should be replaced by 'In this study'. The adjusted text reads as follows:

*"In this study, a rectangular shape is assumed."*

**Point 1.29: Page 9, line 22, 'this study' --> 'Muller (1996)'?**

We appreciate the comment from Referee #1. We agree that the study referred to in this sentence should be indicated by the author and the year of publication. The adjusted sentence reads as follows:

*"The conclusion of Müller (1996) was that the equation proposed by Arneson et al. (2012) in the Hydraulic Engineering Circular No. 18 (HEC-18) was suitable for quantifying the scour depth."*

**Point 1.30: Page 9, line 24, 'the magnitude of the scour' --> 'the scour depth'?**

We agree that the 'magnitude of the scour' should be replaced by 'the scour depth', because in this study the scour is directly measured through the scour depth. The sentence was adjusted as follows:

*"The conclusion of Müller (1996) was that the equation proposed by Arneson et al. (2012) in the Hydraulic Engineering Circular No. 18 (HEC-18) was suitable for quantifying the scour depth."*

**Point 1.31: Page 9, line 27, The meaning subscript of Y is not clear. (c-d, e-d, c-o, e-o, etc.) Maybe you can specify this by a figure.**

We noted that the subscripts of Y (c-d, e-d, c-o, e-o, etc.) could be confusing for the reader. Thus, we decided to change them to the initial letters of demand (d), offer (s), pier (p) and abutment (a). The new subscripts of Y and their meanings are:

$Y_d$: scour demand

$Y_s$: scour supply

$Y_{d-p}$: scour demand on piers

$Y_{d-a}$: scour demand on abutments

$Y_{s-p}$: scour supply on piers

$Y_{s-a}$: scour supply on abutments

**Point 1.32: Page 9, line 27, Eq. (8) and (9) --> Eqs. (8) and (9).**

We appreciate the comment from Referee #1. Indeed, the term 'Eq.' should be plural ('Eqs.') because it refers to both equations (8 and 9). The corrected sentence reads as follows:

*"To incorporate the debris accumulation, the scour demand on the piers ($Y_{d-p}$) is modelled*
5  *with Eqs. (8) and (9) of the NCHRP (2010)."*

**Point 1.33: Page 11, Can you define h_design, e_super, h_imp etc. using figure?**

We sincerely appreciate the comment from Referee #1. In order to better describe some geometric variables of the system, the main variables ($h_{lahar}$, $e_{super}$, $h_{imp}$, $Y_s$, $Y_d$) were defined using Figure 1, as suggested by Referee #1. The meaning of these variables is described in
10  the paragraph located under Figure 1 and in Table 1. Figure 1 was adjusted as follows:

[Figure]

***Figure 1:*** *Free-body diagram of bridge resisting and demanding forces and moments in the presence of a lahar.*

**Point 1.34: Page 11, 'grava'-->'gravel'**

15  We appreciate the suggestion of Referee #1. The variables '$\gamma_{Grava}$' and '$D_{Grava}$' in Eq. (19) should be '$\gamma_{Gravel}$' and '$D_{Gravel}$'. The corrected Eq. (19) reads as follows:

$$M_i = \begin{cases} v_{Lahar} \sqrt{\gamma_{Gravel} \frac{4}{3} \pi \left(\frac{D_{Gravel}}{2}\right)^3} \left(h_{imp} + Y_s\right) & h_{imp} \leq h_{Design} \\ 0 & h_{imp} > h_{Design} \end{cases}, \qquad (19)$$

20  **Point 1.35: Table 1, Definition of numbers in Deterministic value/probabilistic distribution is not clear. E.g. what is the meaning of 1.0 and 1.9 for $K_{gr}$? What 1.0, 1.06 and 1.06 mean for $K_R$? What is the meaning of '1,1' for $K_3$?**

We sincerely appreciate the comment from Referee #1. In the fourth column of Table 1 the deterministic value or probabilistic distribution of the system basic variables are detailed. If

the variable is stochastic, the probabilistic distribution and the related parameters are indicated. If the variable is deterministic, the constant value of the variable is indicated.

For example, $K_{gr}$ has a uniform distribution with parameters 1.0 and 1.9 (i.e. minimum 1.0 and maximum 1.9). $K_R$ has a triangular distribution with parameters 1.0, 1.6 and 1.6 (i.e. minimum 1.0, maximum 1.6 and mode 1.6). The value of $K_3$ was incorrect. It should be '1.1' instead of '1,1', because it is a deterministic variable.

**Point 1.36:** **Table 2, can you show the type of failure (actual and predicted) for each bridge? Also can you show the predicted failure probability for each bridge?**

Regrettably, there is a lack of quality empirical information related to the impacts of volcanic events on Chilean bridges. From the agencies reports we can only know if the bridges failed or not due to the lahars. Thus, we cannot know in detail what was the type of failure (pier/abutment overturning and/or deck sliding).

Once again, the authors appreciate the comments made by Referee #1 and believe that the manuscript improved significantly after including the suggested adjustments.

**REPLY TO REFEREES AND GUIDE TO THE REVISION OF THE PAPER**

**Natural Hazards and Earth System Sciences**

**Title:** Development of Bridge Failure Model and Fragility Curves for Infrastructure Overturning and Deck Sliding due to Lahars

**Authors:** Joaquín Dagá, Alondra Chamorro, Hernán de Solminihac, Tomás Echaveguren

**MS Nº:** nhess-2017-330

**Anonymous Referee #2**

The authors appreciate the comments made by Referee # 2. In this version of the paper, the text, figures, tables and equations were adjusted taking into account all the suggestions of the referees. In addition, the writing, punctuation and English level were improved.

**Point 2.1:** **Section 6.2, and all references to the statistical validation (p. 1, lines 27-29; section 6.3 first paragraph; p. 22, line 16) should be removed completely.**

We sincerely appreciate the comment from Referee #2. Indeed, the number of empirical data for the fragility curves validation (section 6.2) is too small. Thus, the probability of not rejecting the null hypothesis when it is actually false is high due to the small sample.

There is a lack of quality empirical information related to the impacts of volcanic events on infrastructure. Although we used all the records of lahars produced during the eruptions of the Villarrica volcano in 1964, 1971 and 2015, and the Calbuco volcano in 1961 and 2015, which were the most destructive volcanic events in Chile in the last 50 years, there is not enough empirical data available.

We adopted the suggestion of Referee #2 and removed the section 6.2 (and all the related references) and the section 6.3 first paragraph. The adjusted paper has a section '6 Evaluation of the bridge failure models against empirical data and analysis of results', in which the validation of the failure model is explained and analyzed considering the 15 empirical data. In addition, the lack of empirical information for the validation of fragility curves is highlighted. The results of the failure models and the developed fragility curves are also analyzed.

**Point 2.2:** **In section 6.3, the authors should additionally highlight the data-deficiency issue (possibly as a source of future work) and critically evaluate the success of the model in reference to Table 2. How representative is the empirical data of the range of conditions used to generate the analytical fragility curves? Are the main sources of force/vulnerability sufficiently explored with the empirical data and/or could qualitative 'bounds' on the reliability of these curves be determined?**

Again we appreciate the comment from Referee #2. In the new section 6 and in 'Conclusions' the deficiency and lack of empirical data to statistically validate the developed fragility curves is highlighted:

*"However, there is insufficient empirical data to provide a statistical validation of the bridge fragility curves. There are only 15 empirical points ($h_{Lahar}, p_e$) to validate two fragility curves (C1 bridges and C2 bridges). Thus, a deficiency of empirical data on impacts of lahars on bridges is identified."*

In addition, section '7 Conclusions and recommendations' indicates that the statistical validation of developed fragility curves using empirical data is a source of future research:

*"In addition, the empirical data deficiency demonstrates the need to develop more effective protocols to report damage from volcanic events on bridges. With this, the empirical validation of developed fragility curves is a source of future research."*

Finally, in section 6 the ranges of the empirical values of some variables (lahar density, lahar speed, slope, bridge length, bridge width, etc.) of the system lahars-bridges of the 15 historical data are given and analyzed:

*"The historical data of Table 2 consider lahars from 1.5 m to 5.0 m of depth, covering the entire range of hazard intensity of developed fragility curves (1.5 m to 4.0 m). The density of the evaluated lahars ranges from 16,000 to 19,000 $N/m^3$; the slope from 1.0° to 1.2°; the bridge length from 11.3 m to 72.5 m; the bridge width from 3.9 m to 8.3 m; the bridge height from 2.5 m to 5.5 m; the number of deck support from 0 to 5; the bridge height from 2.5 m to 8.3 m; the number of deck support from 0 to 5; the bridge materials are concrete and wood; the number of bridge lanes are 1 and 2. Thus, the empirical data evaluated demonstrate representativeness of the range of the basic variables of the analytical model (Table 1)."*

**Point 2.3: p.1 Line 29: "...that were reached..." change to "affected"**

We agree that the term 'reached' should be replaced by 'affected'. We adopted the suggestion and the corrected sentence reads as follows:

*"Bridge failure models are empirically evaluated using data of 15 bridges that were affected by lahars in the last 50 years."*

**Point 2.4: p.2 Line 6: This implies a lower hazard only, not hazard intensity. Remove "intensity".**

We completely agree with Referee #2. The low probability of occurrence and the small influence area of lava and pyroclastic flows imply only a lower hazard and exposure, but not necessary a lower hazard intensity. A lower risk of lava and pyroclastic flows on road infrastructure is consequently expected. We adjusted the text as follows:

*"Lava and pyroclastic flows may destroy road infrastructure, however, the probability of occurrence of these events is low and exposed areas are limited (Wilson et al., 2014). Considering that risk is a function of the hazard, exposure and vulnerability (UNISDR, 2009), a lower risk of lava and pyroclastic flows on road infrastructure is consequently expected."*

**Point 2.5: p.3 Line 6: "...experimental design was elaborated..." not sure what this means here - possibly reword.**

We agree that the term "experimental design' generates confusion. In order to avoid misunderstandings, this sentence was removed and the title of section '4 Experimental design for modelling infrastructure overturning and deck sliding due to lahars' was replaced by '4 Proposal for modelling substructure overturning and deck sliding due to lahars'. In this section the physical models integrated in the bridge failure model are detailed and the values of the basic variables are indicated.

**Point 2.6: p.6 Figure 1: What is Ft?**

We appreciate the comment from Referee #2. The variable Ft was not defined under Figure 1. The following sentence was added to the paragraph located below Figure 1:

*"The tangential force $F_t$ corresponds to the sum of the hydrodynamic force and the debris impact force applied to the superstructure."*

**Point 2.7: p.9 Line 10: Change to "...a rectangular shape is assumed."**

Again we agree with Referee #2. The term 'rectangular flow' should be replaced by 'rectangular shape'. The corrected sentence reads as follows:

*"In this study, a rectangular shape is assumed."*

**Point 2.8: p.10 Equation 14: Change to Yw;found (misspelt as fuond)**

We appreciate the comment from Referee #2. The variable '$y_{w,fuond}$' in Eq. (14) should be '$y_{w,found}$'. The adjusted Eq. (14) reads as follows:

$$y_{w,found} = Y_s - \frac{Y_d}{3}, \tag{14}$$

**Point 2.9: Equations 13, 20, 21, 24 use the parameter L, which is bridge width. Introduce it at the first instance (Eq. 13) and I would suggest changing it to a less confusing variable (perhaps T for thickness).**

We completely agree that the use of the parameter 'L' for the bridge width could be confusing for the reader. To avoid this confusion, the 'L' parameter was changed to 'T' (Thickness) in the Eqs. 13, 20, 21, 22, 23, 24 and in Table 1, as suggested by Referee # 2.

In addition, the variable 'T' was defined before the Eq. 13, as shown below:

*"The resulting hydrodynamic force exerted by the lahar on the foundation ($F_{w,found}$) and the height at which this force acts with respect to the turning axis ($y_{w,found}$) are given by Eq. (13) and Eq. (14), where the variable T corresponds to the bridge width:"*

$$F_{w,found} = T C_D \left(\frac{\gamma_{Lahar}}{2g}\right) v_{Lahar}^2 \left(\frac{Y_d^2}{h_{Lahar} + Y_d}\right), \tag{13}$$

**Point 2.10: p.13 Table 1: Variable esuper is not listed on the table - what is its value?**

We sincerely appreciate the comment from Referee #2. Indeed, the variable '$e_{super}$' was not listed on Table 1. To show the name, unit, value (probabilistic distribution) and reference of '$e_{super}$', the following row was added in Table 1:

***Table 1:*** *Basic variables involved in the limit state functions.*

| Variable | Name | Unit | Deterministic Value/ Probabilistic Distribution | Value Reference |
|---|---|---|---|---|
| $e_{Super}$ | *Superstructure Thickness* | *cm* | *Gen. Ext. Value (18.6; 4.7; 0.3)* | *Bridge Inventory (MOP)* |

**Point 2.11: p.16 Line 20: This is a valid statement for your study, although I believe it may vary with type and depth of foundations (e.g. the use of piers).**

We completely agree with Referee #2 that we can indicate that deck sliding is not a triggering factor of bridge failures only in this case (Villarrica and Calbuco volcanoes). In order to highlight this, the text was adjusted as follows:

*"The deck sliding is not a triggering factor of bridge failures due to lahars generated by Villarrica and Calbuco volcanoes."*

**Point 2.12: p.17 Line 29: Remove "...it was concluded that its..."**

We appreciate the comment from Referee #2. We adopted the suggestion and removed the term 'it was concluded that its'. The adjusted sentence reads as follows:

*"Regarding the bridges with two or more spans (C2), it was concluded that its collapse height due to lahars could be represented by a cumulative lognormal distribution with µ equal to 0.63 and β equal to 0.13."*

Once again, the authors appreciate the comments made by Referee #2 and believe his/her suggestions and observations have greatly improved the manuscript.